# Efficient Sketches for Training Data Attribution and Studying the Loss Landscape

**Andrea Schioppa**
Google DeepMind
Amsterdam, the Netherlands
`arischioppa@google.com`

## Abstract

The study of modern machine learning models often necessitates storing vast quantities of gradients or Hessian vector products (HVPs). Traditional sketching methods struggle to scale under these memory constraints. We present a novel framework for scalable gradient and HVP sketching, tailored for modern hardware. We provide theoretical guarantees and demonstrate the power of our methods in applications like training data attribution, Hessian spectrum analysis, and intrinsic dimension computation for pre-trained language models. Our work sheds new light on the behavior of pre-trained language models, challenging assumptions about their intrinsic dimensionality and Hessian properties.

## 1 Introduction

**Overview** In this work, we investigate gradient and Hessian vector product (HVP) sketching to address the memory constraints inherent in applications that require the storage of numerous such vectors. Examples of such applications include training data attribution (TDA), eigenvalue estimation, and the computation of the intrinsic dimension. While previous studies on the intrinsic dimension have employed the Fastfood Transform to mitigate memory demands, we demonstrate its theoretical limitations for sketching and its persistent memory bottlenecks on modern accelerators. To resolve these issues, we propose novel sketching algorithms designed for modern hardware and underpinned by robust theoretical guarantees. Our experiments on pre-trained language models demonstrate the scalability of our methods while offering new perspectives on the intrinsic dimension and the Hessian of generative language models.

**Motivation** Training data attribution (TDA) [20, 11] and Hessian eigenvalue estimation [12] offer powerful insights into neural network behavior. TDA requires storing vectors of the same dimensionality $N$ as the network's parameters for each training point, scaling linearly with dataset size ($O(NT)$ for $T$ training points). Similarly, numerically stable Hessian eigenvalue estimation algorithms demand repeated Hessian-vector product (HVP) computations and storage, scaling with network size and iterations ($O(NT)$ for $T$ iterations). These memory bottlenecks hinder the study of large-scale models; to address this, sketching [27] provides a compelling solution. By projecting gradients or HVPs into lower-dimensional random subspaces, sketching preserves their essential geometric properties while drastically reducing memory requirements. However, in TDA sketching has been employed with limited effectiveness. Random projections have been carried out with dense matrices, that introduce significant scaling constraints ($O(ND)$ memory for a target dimension $D$) which necessitate the restriction of gradients to a subset of layers. Current TDA scaling methods require layer selection, as demonstrated with BERT in [8], where only 15M parameters were used out of 110M. Beyond the computational cost of dense matrices, our experiments show that layer selection introduces substantial distortion in the estimation of TDA scores (Sec. 5.2). Recent work on neural network geometry [14] suggests using the Fastfood Transform for gradient sketching

38th Conference on Neural Information Processing Systems (NeurIPS 2024).

(abbr. **FFD** [13]) with $O(N)$ memory; however, as the Fastfood Transform is a random feature generation algorithm, it necessitates indirect application in order to sketch gradients. This has both theoretical and practical consequences. On the one hand, our theoretical analysis reveals that **FFD** fails to fully satisfy the algorithmic requirements of sketching (Thm. 3.1). On the other hand, our experiments (Tab. 3) demonstrate that **FFD** exhibits unacceptable run-time performance on TPUs. These limitations, both theoretical and practical, underscore the need for novel sketching algorithms. To this end, we investigate new sketching paradigms optimized for modern accelerators like GPUs and TPUs. Our work makes the following contributions:

1. *Scalable Gradient Sketching for Modern Accelerators and Networks*: We introduce algorithms (**AFFD**, **AFJL**, **QK**) (Sec. 3) designed to overcome performance limitations of existing sketching techniques with modern neural networks on architectures like GPUs and TPUs. Our design analysis provides further insights into the effectiveness of our approach.

2. *Robust Theoretical Foundations*: We establish theoretical guarantees for **AFFD** and **QK** (Thms. 3.2, 3.3) for sketching and demonstrate limitations in the theoretical basis of the Fastfood Transform (Thm. 3.1). Our analysis further indicates a dimensionality reduction advantage for **AFFD** over **QK**, a finding supported by our TDA experimental results (Sec. 5.3).

3. *Algorithmic Improvements*: We propose more efficient algorithms for the intrinsic dimension estimation and Hessian eigenvalue computation (Sec. 4).

We demonstrate how our methods enable large-scale applications in training data attribution, intrinsic dimension computation, and Hessian spectra analysis with pre-trained language models. This leads to the following insights that advance the understanding of pre-trained language models:

1. *Limitations of Layer Selection*: We demonstrate that layer selection methods yield inaccurate influence score estimations in training data attribution, so their usage should be avoided (Sec. 5.2).

2. *High Intrinsic Dimension*: In contrast to assumptions drawn from classification-based studies, we demonstrate that the intrinsic dimension of LLMs can approach their full parameter count (Sec. 5.4). This challenges prevailing beliefs about the intrinsic dimensionality of these models [14, 1, 17].

3. *LLM Hessian Spectra*: Our analysis shows distinct characteristics of LLM Hessian spectra (Sec. 5.5), contrasting with conjectures based on findings related to smaller networks [9, 21, 3, 12].

**Paper organization** Sec. 2 provides a survey of relevant research in the field, contextualizing our contributions. Sec. 3 introduces our novel sketching algorithms. We begin with necessary background material, analyze design choices, and provide a step-by-step implementation tutorial in Appendix B. Sec. 4 outlines our proposed techniques for efficient intrinsic dimension search and Hessian eigenvalue computation. Sec. 5 describes our experimental setup: subsections are aligned with Sections 3 and 4 to enhance the connection between theory and empirical results

## 2   Related Work

**Sketching**   Sketching algorithms have been extensively studied (see surveys [27, 18, 15]). Our algorithms, **AFFD** and **AFJL**, draw inspiration from the seminal **FJL** algorithm [2] and the **FFD** approach [13]. However, these techniques were designed before the era of modern accelerators like GPUs and TPUs. Therefore, our work revisits their design, optimizing them for modern neural networks and hardware. Theoretically, while [2] established **FJL** as a sketching algorithm, their proof relies on independence assumptions that do not hold in our setting. To address this, we employ more sophisticated concentration tools tailored to bi-linear forms (Thm. 3.2) and the special orthogonal group (Thm. 3.3). Recent work on PAC bounds [17] leveraged Kronecker-product decompositions to accelerate gradient sketching when computing intrinsic dimensionality. Our **QK** algorithm extends the concepts introduced in [17]. Importantly, we provide a proof that **QK** is a sketching algorithm, absent in [17]. Additionally, we demonstrate that the Kronecker structure used in [17] is not essential for performance gains, highlighting that the true bottleneck in **FFD** and **FJL** is memory access. For a comprehensive comparison with [17], please refer to Appendix C. Finally, to support our eigenvalue estimation, we leverage the theoretical guarantees outlined in [25].

**Intrinsic dimension** The concept of intrinsic dimension (ID) offers a valuable metric for understanding the complexity of learning tasks. Originally employed to analyze loss landscapes [14], intrinsic dimension has expanded into the study of language models. [1] demonstrated its role in explaining the generalization power of fine-tuned pre-trained language models. However, their focus was limited to classification tasks. In contrast, our work extends the analysis of ID to generative tasks. This distinction is crucial as we identify scenarios where the task's intrinsic dimension approaches the full model size, a phenomenon not typically observed in classification settings. Additionally, while **FFD** has been used for efficient language model fine-tuning [16], memory constraints limited their investigation of the intrinsic dimension (ID) in generative tasks to 500k; in other words, because of scalability contraints, [16] could only work with a target sketching dimension $\leq$ 500k, which prevented searching for the true value of ID as we demonstrate on a summarization task where ID approaches the model dimension. Therefore, our work overcomes this limitation, allowing us to compute the intrinsic dimension of such tasks.

**Scaling up influence functions** Scaling influence functions for training-data attribution remains a crucial research direction. Previous works like [8, 7] have improved index creation, retrieval speed, and Hessian estimation. However, these approaches can still be computationally demanding. Our work takes a distinct path, aiming to make influence function calculations fundamentally more efficient. Our HVP sketching methods seamlessly replace existing HVP and gradient computations in frameworks proposed within [8, 7, 20]. Furthermore, we offer eigenvector sketching to enhance methods like the Arnoldi iteration [24]. [19] has proposed to use dense random projections by materializing them in chunks and on-the-fly; drawbacks of this approach are: (1) the lack of the scalability in the target sketching dimension and (2) the need of hardware-dependent custom implementations; on the other hand, our approach removes the requirement for specialized implementations (e.g., custom CUDA kernels). This flexibility enables easy integration into standard ML workflows using higher-level languages such as Jax. An orthogonal direction to scaling up via sketching is that of using surrogate models, compare [5].

**Hessian evolution during training** Investigating how the Hessian evolves during training has shed light on the dynamics of deep learning, with seminal works [21, 9] offering intriguing findings. These studies suggest the progressive disappearance of negative eigenvalues and the confinement of gradient descent within a small subspace. While [12] developed a numerically stable Hessian analysis algorithm, its computational demands hinder its application to large-scale models over numerous iterations. Our work addresses this limitation by introducing sketching techniques to enable the efficient construction of large Krylov subspaces (e.g., $10^3$-dimensional) for models like GPT-2L (770M parameters). This advancement significantly surpasses the memory constraints of the method utilized by [12]: a 3TB fp32 storage requirement would have been necessary for a comparable analysis using their approach. Consequently, we are uniquely positioned to rigorously examine the conjectures proposed [21, 9] within the context of pre-trained language models.

## 3 Design Principles for Efficient Sketching Algorithms

In this section, we explore a design space for more efficient sketching algorithms. To establish a foundation, we first analyze the performance bottlenecks of existing algorithms, specifically **FJL** and **FFD**, within the context of modern accelerators. This examination highlights two critical design choices: whether the gradient is sketched implicitly or explicitly, and the kind of pre-conditioner that is used. Informed by this analysis, we propose three novel algorithms: **AFFD**, **AFJL**, and **QK**. We then delve into the theoretical underpinnings of **AFFD** and **QK**, providing rigorous proofs for their guarantees. Additionally, we demonstrate that **FFD** lacks the theoretical foundation required for sketching. Relevant experimental findings are presented in Sections 5.2 and 5.3.

**Dense sketches and FJL** A $D$-dimensional sketch of the gradient of a real-valued function $L(\theta)$ ($\theta \in \mathbb{R}^N$) is a random projection of the gradient $\nabla L \in \mathbb{R}^N$ to $\mathbb{R}^D$. To ensure this projection preserves essential geometric properties, the random projection operator $\Phi : \mathbb{R}^N \to \mathbb{R}^D$ must, with high probability, concentrate the norm of $\Phi(x)$ around the norm of $x$. Mathematically, for each $\varepsilon$ and $\delta$ there exists a large enough target dimension $D(\varepsilon, \delta)$ so that for $D \geq D(\varepsilon, \delta)$:

$$\text{Prob}\left( |\|\Phi(x)\|_2 - \|x\|_2| \geq \varepsilon \|x\|_2 \right) \leq \delta. \tag{1}$$

This concept generalizes to sketching higher-order derivatives (see Appendix C). For our purposes, consider the Hessian vector product operator $\text{HVP} : \mathbb{R}^N \to \mathbb{R}^N$ defined as $\text{HVP}(u) = \nabla^2 L(\theta)(u)$ A sketch of the HVP can be obtained as $v \mapsto \Phi(\text{HVP}(\Phi^T v))$, $v \in \mathbb{R}^D$. This sketch defines a linear mapping $\mathbb{R}^D \to \mathbb{R}^D$ [25]. While a simple **Dense Sketch** (using a random $D \times N$ Gaussian matrix) ensures the norm property, it has $O(DN)$ memory and $O(DN^2)$ compute requirements. The **FJL** algorithm [2] addresses the compute cost with a sparser projection matrix:

$$\Phi(x) = \sqrt{\frac{N}{D}} \cdot G_s \cdot H_N \cdot B(x), \tag{2}$$

where: $B \in \mathbb{R}^{N \times N}$ is a diagonal matrix with $B_{i,i} = \pm 1$ (random signs); $H_N$ is the $N$-dimensional Walsh-Hadamard transform (Appendix C); $G_s$ is a sparse $\mathbb{R}^{D \times N}$ Gaussian matrix with (in-expectation) $\Theta(D \log^2 M)$ non-zero entries ($M$ is a parameter). The $H_N \cdot B$ component preconditions sparse inputs $x$ for which (1) might otherwise fail. Implementing $G_s$ efficiently presents challenges on modern hardware and frameworks like Jax that lack native sparsity support.

**FFD: Implicit Gradient sketching**    The **FFD** transform, introduced by [13] in the context of kernel machines, provides a computationally efficient way to approximate high-dimensional feature maps. As a random feature generator [15], **FFD** constructs high-dimensional random features ($\in \mathbb{R}^N$) from a lower-dimensional input vector ($u \in \mathbb{R}^D$). Given $u \in \mathbb{R}^D$, **FFD** concatenates $\frac{N}{D}$ vectors of the form:

$$\Phi_i(u) = \sigma_F \cdot H_D \cdot G_v \cdot \Pi \cdot H_D \cdot B(u) \quad (1 \le i \le \frac{N}{D}), \tag{3}$$

where $B$ and $H_D$ are as in (2) (and are both $D \times D$-matrices), $\Pi$ is a permutation matrix, $G_v$ is a diagonal $D \times D$-matrix with i.i.d. standard Gaussian entries, and $\sigma_F$ is a normalization constant (see [13]; for a practical implementation see the code released by [16]). **FFD** has the key advantage of constant memory cost, $O(N)$, regardless of the input dimension $D$. Since **FFD** defines a map $\mathbb{R}^D \to \mathbb{R}^N$, direct sketching of a gradient is not possible. To address this, [14] perform what we call an *Implicit Gradient Sketch*:

$$\mathcal{S}(\nabla_{\theta|\theta_0} L) = \nabla_{\omega|0} L(\theta_0 + \textbf{FFD}(\omega)); \tag{4}$$

this formulation effectively applies the transpose of **FFD** to the gradient. While [13] establish certain properties of **FFD**, a complete proof of its suitability as a random feature generator is missing. Additionally, whether the **FFD** satisfies sketching guarantees (1), remains to be fully investigated (we will address it in Thm. 3.1).

**Explicit sketches.**    In light of equation (4), it's natural to consider whether a direct sketch of the gradient could be achieved using a map $\Phi : \mathbb{R}^N \to \mathbb{R}^D$. We define this as an *Explicit Gradient Sketch*:

$$\mathcal{S}(\nabla_{\theta|\theta_0} L) = \Phi(\nabla_{\theta|\theta_0} L). \tag{5}$$

This approach offers flexibility. For random feature generation methods like **FFD**, $\Phi$ needs to be implemented as the transpose; for sketching algorithms like **FJL**, the explicit sketch can be applied directly, while the transpose would be needed for the implicit form (4). In Appendix B, we provide a Jax-based tutorial on transposing sketching algorithms. Which one is the right approach? Intuitively, implicit sketches may seem more efficient since they avoid direct gradient materialization. However, as we'll demonstrate in Section 5.3, explicit sketches surprisingly offer significant performance advantages. Table 4 quantifies the substantial wall-time reductions (approximately 70% on average) across various algorithms when using explicit sketches.

**Removing the lookup bottleneck.**    In both **FJL** and **FFD** algorithms, multiplications by $G_s$ and $\Pi$ introduce a lookup-based memory bottleneck on modern accelerators. This hinders performance, as seen in unacceptable early TPU results (compare Tab. 3). To address this, we propose randomizing the pre-conditioner $H_N$. Efficient implementations of $H_N$ leverage the fact that it can be decomposed using Kronecker products; specifically, $H_{AB} = H_A \otimes H_B$, which allows a recursive multiplication by $H_N$ in $O(N \log N)$-time and $O(N)$ storage. We exploit this by permuting rows/columns within Kronecker factors, reducing memory access costs from $O(AB)$ to $O(A+B)$ in the previous example. For optimal accelerator usage, we limit factors to roughly 1024 columns. Building on this, we

modify (3). In the resulting **AFFD** algorithm (6), we remove $\Pi$, use row-permuted factors in $H_N^{\pi_1}$, and column-permuted factors in $H_N^{\pi_2}$:

$$\Phi(x) = R_D(\sqrt{\frac{N}{D}} \cdot H_N^{\pi_2} \cdot G_v \cdot H_N^{\pi_1} \cdot B(x)), \tag{6}$$

where $R_D$ denotes restriction to the first $D$ coordinates. While all matrices are now $N \times N$, efficient Hadamard implementations avoid large matrix materializations (see Appendix B for our Jax code). We further introduce **AFJL** (7), where $H_N^{\pi_2}$ is removed:

$$\Phi(x) = R_D(\sqrt{\frac{N}{D}} \cdot G_v \cdot H_N^{\pi_1} \cdot B(x)). \tag{7}$$

This can be seen as replacing $G_s$ in **FJL** with a diagonal Gaussian matrix. Generalizations with multiple $(G_v, H_N^{\pi_1})$ pairs are possible but weren't explored due to the vanilla version's strong results. Empirical results on TPUs (Table 3) show the crucial impact of our changes. **AFJL** achieves a 100x wall-time reduction over **FJL**, **AFFD** a 64x reduction over **FFD**. While GPU wall-time remains similar, peak memory usage significantly improves. Appendix A highlights **FJL**'s scaling issues beyond $D = 2^{20}$ on GPUs.

**Alternative pre-conditioners.** To address the smoothing of sparse inputs in sketching algorithms, [2] introduced $H_N$ in (2). The theoretical basis lies in the Heisenberg uncertainty principle, leveraging the Fourier Transform on a discrete group (see [2]). However, the computationally efficient Fast Fourier Transform (FFT) shares this desirable property. This raises the question of whether the FFT might yield performance gains for sketching. We find that replacing $H_N$ with the FFT offers significant advantages. Experimentally, it reduces wall time by 62% on GPUs (Tab 4). Inspired by the Kronecker product structure enabling efficient $H_N$ implementation, we propose another generalized pre-conditioner, $Q$. This random $N \times N$ orthogonal matrix has a Kronecker decomposition of $K$ independent orthogonal matrices of sizes $\{B_i \times B_i\}_{i=1}^K$, sampled according to the Haar measure on $SO(B_i)$. Our approach allows direct application of $Q$ without the additional diagonal matrix $B$, and offers a 40% wall time reduction on GPUs (Table 4). This $Q$ pre-conditioner has the potential to unlock broader optimizations within sketching algorithm design as discussed in the next subsection.

**Direct usage of the pre-conditioner $Q$.** Inspired by the design of the pre-conditioner $Q$, we introduce a novel sketching algorithm, **QK**. This ablation explores the direct use of $Q$ to transform the input, potentially leading to improved efficiency and memory usage. **QK** is defined as:

$$\Phi(x) = \sqrt{\frac{N}{D}} \cdot Q(x). \tag{8}$$

Here, $Q$ is a random $D \times N$-orthogonal matrix with a Kronecker product decomposition of $K$ independent orthogonal matrices of sizes $\{D_i \times B_i\}_{i=1}^K$. Each factor is generated by sampling from $SO(B_i)$ according to the Haar measure and restricting to the first $D_i$ rows. Importantly, **QK** generalizes the approach of [17] by employing more Kronecker factors. This offers the potential for significant memory reductions (Appendix C).

**Diagrams.** Figure 1 illustrates **AFFD**, **AFJL** and **QK** with diagrams.

**Theoretical results** We now delve into the theoretical underpinnings of our proposed algorithms, emphasizing the interplay between theory and experimentation. A key finding is a limitation of the **FFD** algorithm:

**Theorem 3.1.** *There are some inputs $x$ for which **FFD** does not satisfy the sketching property* (1).

Note that this limitation is specific to **FFD** and not inherent to implicit-mode algorithms. This distinction is important: the explicit-mode formulation allowed to simplify the theoretical analysis to prove Thm. 3.1. Next, we establish a theoretical guarantee for our **AFFD** algorithm. Compared to [2] there is added complexity as independence arguments cannot be used for $G_v$; we thus apply the Hanson-Wright inequality for quadratic forms to prove:

**Theorem 3.2.** *AFFD satisfies* (1) *with*

$$\delta = \delta_1 + 2\exp\left(-C\varepsilon^2 \frac{D}{4\log^2 \frac{2N}{\delta_1}}\right), \tag{9}$$

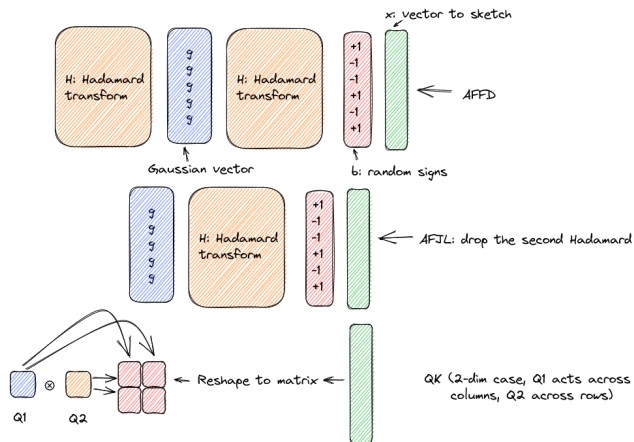

Figure 1: Diagram to illustrate our proposed sketching algorithms.

*for a universal constant C and for any $\delta_1 > 0$.*

Finally, we analyze **QK**. Its structure allows for concentration arguments on orthogonal groups, leading to:

**Theorem 3.3.** *QK satisfies* (1) *with*

$$\delta = 2 \sum_i \exp(-4CD_i((1+\varepsilon)^{1/K} - 1)^2),$$

*for a universal constant C.*

Crucially, the bound in Thm. 3.3 is less favorable than that of Thm. 3.2. This is due to the $1/K$-root, summation over sub-dimensions, and concentration depending on the $D_i$. These theoretical insights support our experimental findings in Section 5.3, where **QK** requires higher target dimensions ($D$) to achieve performance comparable to **AFFD**. Because of space constraints the proofs are included in Appendix C.

## 4    Expanding the Utility of Sketching Algorithms

In this Section we expand the usage of sketching to other applications.

**Improving the search for the intrinsic dimension.**    The intrinsic dimension ($D_{int}$) [14], is the minimum dimension ($D$) of a random subspace ($V$) where SGD yields at least 90% of the full model's performance on a target metric ($\tau^1$). Sketching algorithms, more efficient than FastFood, already accelerate the search for $D_{int}$ [17]. Our memory-efficient algorithms enable us to investigate scenarios where $D_{int}$ may approach the model dimension. For instance, [16] applied **FFD** to fine-tune generative language models but capped the target dimension ($D$) to 500k due to memory limits. We propose a novel search algorithm that estimates $D_{int}$ in a single training run. Current methods rely on multiple runs across potential $D_{int}$ values. To streamline, consider a binary search approach (assuming $D_{int}$ is a power of 2) starting from an initial guess ($D_{min}$) up to the model parameter count ($N$). This would require $\lceil \log_2 \frac{N}{D_{min}} \rceil$ runs. Instead, we propose a single training run where $D$ increases progressively. The heuristic: a fixed computational budget ($c$ steps) should yield an expected improvement of at least $\delta$. We start with $D = D_{min}$. If, after $c$ steps, the target metric's improvement is less than $\delta$ or hasn't reached $\tau_{90}$, we double $D$. This yields an estimate $D^*$ within a factor of 2 of $D_{int}$. A subsequent run with $D = D^*/2$ verifies if this factor can be eliminated. See Appendix B for Python code. In Sec. 5.4, we apply this approach to pre-trained language models on classification and generative tasks. We find that $D_{int} \ll N$ for classification, but for the generative task, $D_{int}$ depends on the choice of $\tau$ and can approach the parameter count $N$. This finding is significant, as it

---

[1]While 90% is common, this is a configurable parameter.

challenges prevailing assumptions about intrinsic dimensionality in such models [14, 1, 17]. These studies primarily focused on classification tasks, obtaining generalization bounds like $O(\sqrt{D_{int}})$ ([1, Eq. 4], [17, Sec. 3]) where $D_{int}$ was presumed much smaller than the model dimension. Our results suggest a need for non-linear projection algorithms to achieve lower intrinsic dimensionality for generative tasks.

**Scaling eigenvalue estimation**  Investigating the Hessian's spectrum and eigenvectors often relies on memory-bound iterative algorithms. The Arnoldi algorithm, as used in [12], requires full re-orthogonalization for stability. This necessitates storing vectors as large as the model itself for each iteration: a constraint that limits the number of estimable eigenvalues (e.g., Krylov subspaces of dimension $\simeq 90$ in [12]). Inspired by the theoretical work of [25], we propose sketching to address this memory bottleneck. With a sketching dimension of $10^6$, we can readily construct 1k-dimensional Krylov subspaces for pre-trained language models like BART, Roberta, and GPT-2L. This represents a breakthrough because the method in [12] would demand an impractical 3TB of storage for GPT-2L alone. Our technique enables exploration of conjectures [21, 12, 9] about Hessian structure in the context of fine-tuning large language models. These conjectures are elaborated in the experimental results on eigenvalue estimation (Sec. 5.5). Crucially, we find that Hessian spectra in these models may deviate significantly from behaviors observed in smaller networks.

# 5  Experiments

To thoroughly evaluate the proposed sketching methods, we present a comprehensive set of experiments. First, we highlight the limitations of existing TDA scaling strategies (Sec. 5.2). Next, we dissect the impact of specific design choices on our sketches (Sec. 5.3). We then introduce and validate an algorithm for intrinsic dimension estimation, enabling computational savings (Sec. 5.4) and show-casing that the intrinsic dimensionality of generative tasks can be large. Finally, we apply our techniques to explore the evolution of the Hessian spectrum during pre-trained language model fine-tuning (Sec. 5.5).

## 5.1  How we define the Training-Data Attribution score.

In TDA there are different ways to measure the similarity score between two examples $x$ and $y$. In our experiments we opt for the TDA score defined as $\nabla_\theta L(\theta, x) \cdot \nabla_\theta L(\theta, z)$ because of its simplicity, allowing us to iterate on multiple algorithms and layer selection schemes, and being a building block for more complicated methods. While high correlation with full gradient dot products may not be the definitive measure of long-term TDA performance [19, 23], it is a practical metric in the short time range and a building block of more computationally intensive methods like TRAK [19]. For example, in the short time range, gradient dot products correlate with loss changes and are relevant to select examples for error correction [23]. Evaluating with the LDS from TRAK would introduce more hyper-parameters and considerable more computation: one would need at least 50 models fine-tuned on different subsets of the data; moreover, TRAK itself relies on accurate gradient sketches, as measured by dot products, as the basic building block as TRAK demonstrates that "preserving inner products to sufficient accuracy results in a gradient-descent system that approximately preserves the same evolution as the one corresponding to model re-training" [19, C.2].

## 5.2  Shortcomings of previous Training-Data Attribution scaling strategies.

Past work [20, 28, 8] tackles the memory bottleneck of Training-Data Attribution by calculating gradients restricted to a specific layer and potentially applying a Dense Sketch. Here, we demonstrate that layer selection distorts both influence scores and eigenvalue estimates, while Dense Sketches exhibit poor scaling characteristics. These findings align closely with the first subsection of Sec. 3. Furthermore, advice on layer selection lacks consistency: [20] promotes the last layer, whereas [28] supports using Token Embeddings in NLP tasks. We demonstrate the distortion caused by layer selection on influence scores (the inner product of two gradients). Considering $2^{12}$ pairs of points $(x, z)$, we compute the Pearson correlation $r$ between the TDA score $\nabla_\theta L(\theta, x) \cdot \nabla_\theta L(\theta, z)$ estimated using a layer-specific gradient and the ground truth based on the full gradient. We adopt the setup of [6]: a generative task fine-tuning GPT-2 on the WikiText-103 dataset (BART and zsRE results in Appendix A). **Our findings indicate the unreliability of layer selection (Table 1)**. Correlations

Table 1: Layer selection results in unreliable estimates for influence scores and eigenvalue estimation. The best correlation with ground truth influence scores does not exceed $90\%$ and is quite low for most layers; the relative error in eigenvalue prediction is always at least $20\%$.

| MODEL | LAYER | R | EIG. ERR. |
|-------|-------|------|------|
| GPT-2 | TOK. EMB. | 0.16 | 0.72 |
| GPT-2 | 1 | 0.75 | 0.24 |
| GPT-2 | 2 | 0.89 | 0.31 |
| GPT-2 | 3 | 0.90 | 0.19 |
| GPT-2 | 4 | 0.89 | 0.24 |
| GPT-2 | 5 | 0.78 | 0.37 |
| GPT-2 | 6 | 0.38 | 0.40 |

Table 2: Dense projections on the layers do not scale; for each layer we report the wall time for the maximum dimension that does not result in an OOM.

| LAYER | GPU WALL (MS) | $k$ (MAX VAL BEFORE OOM). |
|-------|-------|-------|
| TOK. EMB. | 31.2 | 32 |
| LAYER 1 | 25.1 | 128 |
| LAYER 2 | 24.4 | 128 |
| LAYER 3 | 23.9 | 128 |
| LAYER 4 | 23.1 | 128 |
| LAYER 5 | 22.4 | 128 |
| LAYER 6 | 21.7 | 128 |

with ground truth rarely exceed $90\%$ and are significantly lower for most layers. In contrast, **AFJL** achieves a $98\%$ correlation with a compact $D = 2^{13}$. Extending the analysis to Hessian-based influence functions by looking into eigenvalue estimation emphasizes the shortcomings of layer selection. We aim to compute the top 10 eigenvalues of both the full Hessian and those restricted to a chosen layer. Even with potential differences in magnitude and location, layer selection could still be valuable if the true eigenvalues were approximated well by applying an orientation-preserving linear map $\mathbb{R}^1 \to \mathbb{R}^1$ to those computed for a particular layer. However, this is not the case, with the relative mean absolute error at $20\%$ on the best layer (Table 1). Finally, **layer selection coupled with dense projections faces severe scalability limitations**. Setting $D = 2^{12}$ within the same setup highlights this issue. Limited memory forces us to divide the computation into $D/k$ projections to $\mathbb{R}^k$ where $k$ is the smallest power of 2 enabling a dense projection without an Out-of-Memory error. For token embeddings, we find $k = 32$, whereas other layers require $k = 128$ on a V100 GPU. Projecting to $\mathbb{R}^k$ takes $\sim 31ms$ for token embeddings, resulting in a total of $\sim 4s$ to project to $\mathbb{R}^D$. Other layers require $\sim 24ms$ per projection to $\mathbb{R}^k$ and $\sim 0.77s$ to project to $\mathbb{R}^D$, see Table 2 for a per-layer breakdown. Finally, an alternative approach to dense projections is to materialize dense matrices on the fly in chunks [19]; this has two substantial disadvantages: (1) the runtime scales linearly with the target dimension (memory is traded off with compute), and (2) specialized kernels are necessary for efficient implementation, with unclear applicability to TPUs; we include a demonstration of these limitations in Appendix A.

### 5.3 Analyzing the Impact of Design Choices

In this section, we analyze the impact of the design choices presented in Sec. 3 on both sketching quality and performance. Regarding **sketching quality**, we find that small values of $D$ often lead to remarkably accurate reconstructions of influence scores. For TDA scores, achieving correlation $r \geq 0.95$ requires the following dimensions: **FJL**, **FFD**, **AFFD**: $D = 2^{10}$; **AFJL**: $D = 2^{12}$; **QK**: $D = 2^{14}$. To reach $r \geq 0.99$, simply increase each dimension by a factor of 8. While memory limitations prevented sketching HVPs with **FJL** and **FFD** for eigenvalue estimation, the other algorithms scaled well. We achieved a relative mean absolute error below $5\%$ with the following dimensions: **AFFD**: $D = 2^{10}$; **AFJL**: $D = 2^{12}$; **QK**: $D = 2^{13}$. Note that **QK** requires larger $D$, consistent with the theoretical comparisons in Theorems 3.3 and 3.2 (Sec. 3). Regarding **perfomance**, we outline here the key findings and refer the reader to Appendix A for a comprehensive analysis

Table 3: Wall-time $T$ and peak memory usage $M$ comparison on gradient sketches for GPT-2. Removing look-ups is crucial for TPU performance and decreasing GPU memory utilization.

| ALGO | GPU (V100) | | TPU (v2) | |
|------|------------|------|----------|------|
| | $T$ (MS) | $M$ (GB) | $T$ (MS) | $M$ (GB) |
| **FJL** | 123 | 6.9 | 8997 | 3.0 |
| **AFFD** | 205 | 3.2 | 134 | 2.8 |
| **FFD** | 197 | 4.1 | 8694 | 4.3 |
| **AFJL** | 116 | 2.9 | 89 | 2.7 |
| **QK** | 82 | 1.7 | 64 | 1.1 |

Table 4: Speed-ups (ratio $R$ of the slowest wall-time to the fastest one) corresponding to changing a design choice (e.g. implicit to explicit or $H_N$ to the FFT.).

| ALGO | GPU (V100) ($R$) | TPU (v2) ($R$) | ALGO | GPU (V100) ($R$) | TPU (v2) ($R$) |
|------|------------------|----------------|------|------------------|----------------|
| | IMPLICIT $\to$ EXPLICIT | | | $H_N \to$ FFT | |
| **FJL** | 1.96 | 1.20 | **AFFD** | 1.98 | 1.00 |
| **AFFD** | 1.81 | 2.07 | **AFJL** | 1.64 | 1.00 |
| **FFD** | 1.76 | 1.10 | | $H_N \to Q$ | |
| **AFJL** | 1.64 | 1.67 | **AFFD** | 1.45 | 1.00 |
| **QK** | 1.41 | 2.45 | **AFJL** | 1.44 | 1.00 |

across design choices. First, *removing look-ups significantly reduces wall-time on TPUs, while on GPUs it substantially lowers peak memory usage* (reductions of 2.4x for **FJL** to **AFJL**, and 1.3x for **FFD** to **AFFD**), see Table 3. Second, *explicit sketching consistently provides substantial speed-ups* (see Table 4); we conjecture this is due to the fact that implicit sketching results in more memory accesses. Finally ,while modifying the pre-conditioner doesn't affect TPU performance, *it significantly improves GPU performance*, see Table 4 under the headings $H_N \to$ FFT and $H_N \to Q$.

## 5.4 Estimating the intrinsic dimension.

Our experiments evaluate the efficiency and accuracy of our intrinsic dimension estimation algorithm (presented in Sec. 4). We consider two experimental setups: classification, where we fine-tune Roberta on SNLI with accuracy as the target metric; generation, where we fine-tune BART on XSUM for text summarization, using Rouge1 and Rouge2 for evaluation. We employ three projection algorithms: **FFD**, **AFJL** (FFT-based variant), and **QK**. We first **validate algorithm consistency** by demonstrate that the algorithm produces consistent estimates of the intrinsic dimension across multiple runs; we repeat each search experiment with three random seeds obtaining estimates within a factor of 2 (Appendix B). We then **verify that the estimated $D^*$ accurately represents the intrinsic dimension $D_{int}$** by fine-tuning the model in a $D = D^*/2$-dimensional subspace and ensuring that the target metric remains below the $\tau_{90}$ threshold (details in Appendix B). We point out that selecting good search parameters $(c, \delta)$ was relatively straightforward by observing target metric improvement during full fine-tuning . Regarding **computational efficiency, our approach requires significantly fewer steps than binary or brute-force search**. For instance, the XSUM search required only twice the steps of a full fine-tuning run and the same amount of steps for the classification task. We finally look at **how the intrinsic dimension varies between classification and generation tasks**. For classification, our findings are consistent with prior work [1], were all algorithms yield $D_{int} = 2^{13} \ll N$. For generation we first observe that the results are **metric-dependent**: for Rouge1, **FFD** and **QK** estimate $D_{int} = 2^{25}$ while **AFJL** yields $D = 2^{24}$; for Rouge2, $D_{int}$ equals the full model dimension $N \sim 2^{27}$. In both cases, however, the intrinsic dimension is close to the full model dimension: this challenges assumptions about intrinsic dimension in the context of generative tasks. While prior studies [14, 1, 17] focused on classification with generalization bounds of the form $O(\sqrt{D_{int}})$, our results indicate that: **generative tasks may exhibit higher intrinsic dimension and generative models may require new non-linear projections to uncover significantly lower intrinsic dimensions.**

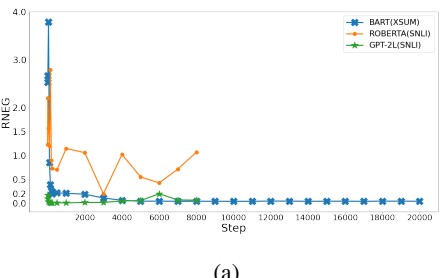
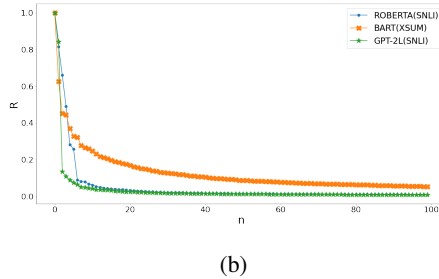

|            (a)            |            (b)            |

Figure 2: **left**: ratio (RNEG) of the absolute value of the top negative to the top positive eigenvalue; **right**: ratio $R$ of the $n$-th largest positive eigenvalue to the largest positive eigenvalue. We define outliers when $R > 20\%$, motivated by [12, Fig.2]. Higher-resolution versions for printing can be found in Appendix A. These results disprove conjectures on the Hessian structure, see Sec. 5.5.

## 5.5 Hessian Analysis During Pre-trained Language Model Fine-Tuning

In this section, we examine the Hessian's evolution during pre-trained language model fine-tuning. Using sketching, as proposed in Section 4, we estimate the Hessian's eigenvalues and eigenvectors. We fine-tune Roberta and GPT-2L on SNLI (3 classes) and BART on XSUM, employing **AFFD**, with a target dimension $D = 2^{20}$, to construct a 1k-dimensional Krylov subspace. This substantially extends the work of [12] who considered smaller models and smaller subspaces (90-dimensional Krylov subspace). Spectrum estimation is performed every 1k steps, including initial steps $\{0, 10, 50, 100, 150, 200, 250, 500\}$. Our goal is to see if observations from previous studies with smaller networks hold in this context:

- **Obs1**: Negative eigenvalues gradually disappear during training [21, 12].
- **Obs2**: $K - 1$ outlier eigenvalues emerge for $K$-class classification, with the gradient aligning to their corresponding subspace [9, 21, 3]. Moreover, these outliers, which hinder optimization, stem from training without normalization [12].

We find that these obervations don't fully translate to pre-trained language model fine-tuning. Regarding **Obs1**, we compute the ratio (RNEG) of the absolute values of the top negative and positive eigenvalues; RNEG shows inconsistent behavior across models (Fig.2 (a)). Roberta maintains a higher RNEG, while GPT-2L and BART see it diminish over time. Regarding **Obs2**, outlier eigenvalues don't strictly adhere to the $K - 1$ rule. Roberta has more outliers (6), and the gradient-outlier alignment is less pronounced (27% Roberta, 35% GPT-2L, 8% BART) compared to smaller networks [12, 9]. See Fig. 2 (b). Moreover, outliers emerge despite layer normalization.

## 6 Conclusions and Limitations

In this work, we have dissected the theoretical and practical limitations of existing gradient sketching techniques when applied to modern neural networks and accelerators. Our analysis motivated the design of novel sketching algorithms, for which we established theoretical guarantees; additionally, we exposed limitations in the theoretical underpinnings of the Fastfood transform. These methods, along with refined intrinsic dimension estimation and Hessian eigenvalue computation, provide an efficient toolkit for model analysis. We successfully apply this toolkit to pre-trained language models, revealing the need to rethink layer-selection-based influence functions, the high intrinsic dimensionality of a generative task, and the deviation of LLMs' Hessian spectra from what observed in smaller networks. While in Sec. 5.4 we exhibit an example of a generative task with a large intrinsic dimension, we leave an in-depth study for future work. We tested the efficiency of our sketching algorithms with Transformers in Sec. 5.3, but results might vary for other model architectures.

## Acknowledgements

We thank Jonathan Heek for helping with optimizing code for TPUs and Katja Filippova for feedback on the draft.

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

# Contents

## A  Appendix: Additional Experimental Results

### A.1  Additional results on layer selection

In Table 5 we include the full results of layer selection of GPT-2 and BART: in Sec. 5.2 we restricted the discussion to GPT-2 because of space constraints; for the purpose of this experiment we consider the setup of [6]: NLP tasks which consists in fine-tuning GPT-2 on the WikiText-103 dataset and BART and zsRE.

### A.2  Quality of sketches for inner products and eigenvalue estimation

For each algorithm, we report in Table 6 the minimal value of $\log_2 D$ necessary to reach a Pearson correlation $> 0.9x$ with the ground truth when estimating inner products of gradients using sketches. As expected from the worse concentration bound, **QK** requires a larger dimension. We conjecture that the fact that **FFD** is effective might be due to the gradient distribution giving $0$-measure to the inputs

Table 5: Most of the time layer selection results in unreliable estimates for influence scores and eigenvalue estimation. The best layer correlation with ground truth influence scores does not exceed $\simeq 90\%$ and is quite low for most layers. The relative error in eigenvalue prediction is always at least $\simeq 20\%$.

| MODEL | LAYER | R | EIG. ERR. |
|---|---|---|---|
| GPT-2 | TOK. EMB. | 0.16 | 0.72 |
| GPT-2 | 1 | 0.75 | 0.24 |
| GPT-2 | 2 | 0.89 | 0.31 |
| GPT-2 | 3 | 0.90 | 0.19 |
| GPT-2 | 4 | 0.89 | 0.24 |
| GPT-2 | 5 | 0.78 | 0.37 |
| GPT-2 | 6 | 0.38 | 0.40 |
| BART | TOK. EMB. | 0.54 | 0.20 |
| BART | DEC. 1 | 0.62 | 0.39 |
| BART | DEC. 2 | 0.88 | 0.43 |
| BART | DEC. 3 | 0.73 | 0.28 |
| BART | DEC. 4 | 0.91 | 0.28 |
| BART | DEC. 5 | 0.84 | 0.19 |
| BART | DEC. 6 | 0.70 | 0.18 |
| BART | ENC. 1 | 0.41 | 0.50 |
| BART | ENC. 2 | 0.45 | 0.59 |
| BART | ENC. 3 | 0.56 | 0.48 |
| BART | ENC. 4 | 0.71 | 0.40 |
| BART | ENC. 5 | 0.91 | 0.46 |
| BART | ENC. 6 | 0.89 | 0.16 |

Table 6: For each algorithm the minimal value of $\log_2 D$ necessary to reach a Pearson $r > x$ where $x = 0.9\{5, 8, 9\}$
for estimating inner products of gradients.

| ALGO | $r > 0.95$ | $r > 0.98$ | $r > 0.99$ |
|---|---|---|---|
| **FJL** | 10 | 12 | 13 |
| **AFFD** | 10 | 12 | 13 |
| **AFJL** | 12 | 13 | 15 |
| **QK** | 14 | 16 | 17 |
| **FFD** | 10 | 12 | 14 |

that **FFD** would fail to sketch (Theorem 3.1). For **AFFD**, **AFJL** and **QK** we report in Table 7 the minimal value of $\log_2 D$ necessary to reach a relative mean absolute error $err < x$ when estimating the top 10 eigenvalues of the Hessian.

## A.3 A closer look at FJL vs AFJL

In Sec. 5.3 we pointed out that the **FJL**'s wall time and memory usage increase with the target dimension $D$. We compare the peak memory usage and the wall time of **FJL** with that of **AFJL** on the inner product task of Sec. 5.3 on GPU (V100), see Figures 3,4: **FJL**'s cost significantly increases with $D$ and does not scale beyond $D = 2^{20}$.

Table 7: For each algorithm the minimal value of $\log_2 D$ necessary to reach a relative error $err < x$ where $x = 0.2, 0.1, 0.05$ in reconstructing the top 10 eigenvalues.

| ALGO | $err < 0.2$ | $err < 0.1$ | $err < 0.05$ |
|---|---|---|---|
| **AFFD** | 9 | 10 | 10 |
| **AFJL** | 12 | 12 | 12 |
| **QK** | 13 | 13 | 13 |

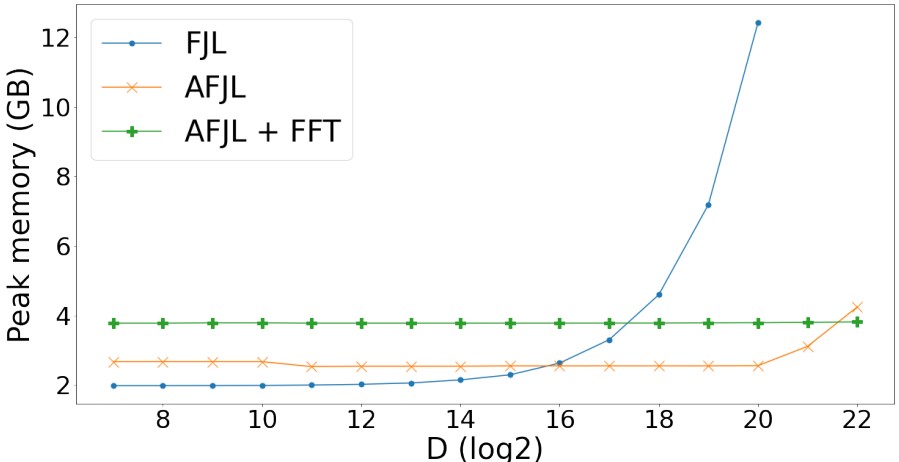

Figure 3: Peak memory usage comparing **FJL** with **AFJL**. Results on GPU (V100); for **FJL** results with $D > 2^{20}$ are not reported as there were Out-of-Memory errors.

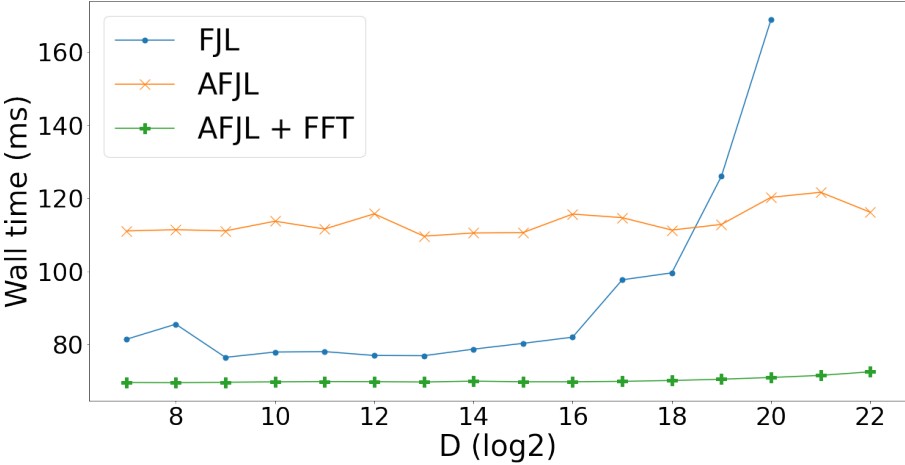

Figure 4: Wall time comparing **FJL** with **AFJL**. Results on GPU (V100); for **FJL** results with $D > 2^{20}$ are not reported as there were Out-of-Memory errors.

## A.4 Compute cost of sketching gradients

In Table 8 we report the compute costs of sketching gradients in the setup of Sec. 5.3.

Table 8: Compute costs of sketching gradients in the setup of Sec. 5.3. I=1 denotes that a method is implicit, F=1 that the FFT is used as a pre-conditioner instead of the Walsh-Hadamard Transform, and Q=1 that random orthogonal matrices decomposed as a Kronecker product are used as a pre-conditioner The lowest time and memory costs are in blue and bold and the highest ones in red and italic. nan indicates a result that was not computed because of XLA compilation errors. For this benchmark we considered a target dimension $D$ in the range $[2^{17}, 2^{22}]$. GPU is V100, TPU is TPUv2.

| ALGO | I | F | Q | GPU WALL (MS) | GPU MEM (GB) | TPU WALL (MS) | TPU MEM (GB) |
|------|---|---|---|---------------|--------------|---------------|--------------|
| **FJL** | 0 | 0 | 0 | 123 | *6.9* | 8997 | 3.0 |
| **FJL** | 0 | 1 | 0 | 85 | 6.1 | 9271 | 4.4 |
| **FJL** | 1 | 0 | 0 | 244 | 6.7 | *10958* | 3.3 |
| **FJL** | 1 | 1 | 0 | 164 | 6.1 | NAN | NAN |
| **AFFD** | 0 | 0 | 0 | 205 | 3.2 | 134 | 2.8 |
| **AFFD** | 0 | 0 | 1 | 142 | 3.2 | 134 | 2.8 |
| **AFFD** | 0 | 1 | 0 | 104 | 4.2 | 134 | 2.8 |
| **AFFD** | 1 | 0 | 0 | *454* | 3.2 | 278 | 3.0 |
| **AFFD** | 1 | 0 | 1 | 237 | 3.2 | 278 | 3.0 |
| **AFFD** | 1 | 1 | 0 | 126 | 4.2 | 278 | 3.0 |
| **FFD** | 0 | 0 | 0 | 197 | 4.1 | 8694 | 4.3 |
| **FFD** | 0 | 1 | 0 | 140 | 5.6 | NAN | NAN |
| **FFD** | 1 | 0 | 0 | 352 | 4.2 | 9600 | *4.7* |
| **FFD** | 1 | 1 | 0 | 239 | 4.8 | NAN | NAN |
| **AFJL** | 0 | 0 | 0 | 116 | 2.9 | 89 | 2.7 |
| **AFJL** | 0 | 0 | 1 | 81 | 2.9 | 89 | 2.7 |
| **AFJL** | 0 | 1 | 0 | **71** | 3.8 | 89 | 2.7 |
| **AFJL** | 1 | 0 | 0 | 231 | 3.0 | 149 | 2.8 |
| **AFJL** | 1 | 0 | 1 | 121 | 3.1 | 149 | 2.8 |
| **AFJL** | 1 | 1 | 0 | 88 | 4.1 | 149 | 2.8 |
| **QK** | 0 | 0 | 1 | 82 | **1.7** | **64** | **1.1** |
| **QK** | 1 | 0 | 1 | 116 | 1.7 | 157 | 1.4 |

## A.5   Compute cost of sketching HVPs

In Table 9 we report the compute costs of sketching HVPs in the setup of Sec. 5.3.

Table 9: Compute costs of HVP using sketching. I=1 denotes that a method is implicit, F=1 that the Fourier Transform is used instead of the Walsh-Hadamard Transform and Q=1 that random orthogonal matrices decomposed as a Kronecker product are used instead of Hadamard matrices. The lowest time and memory costs are in blue and bold and the highest ones in red and italic. GPU is V100, TPU is TPUv2.

| ALGO | I | F | Q | GPU WALL (MS) | GPU MEM (GB) | TPU WALL (MS) | TPU MEM (GB) |
|------|---|---|---|---------------|--------------|---------------|--------------|
| **AFFD** | 0 | 0 | 0 | 397 | 2.8 | 158 | 2.7 |
| **AFFD** | 0 | 0 | 1 | 183 | 2.8 | 158 | 2.7 |
| **AFFD** | 0 | 1 | 0 | 111 | 3.3 | 158 | 2.7 |
| **AFFD** | 1 | 0 | 0 | *627* | 3.1 | *268* | 3.6 |
| **AFFD** | 1 | 0 | 1 | 238 | 3.1 | 267 | 3.6 |
| **AFFD** | 1 | 1 | 0 | 137 | 3.7 | 268 | *3.6* |
| **AFJL** | 0 | 0 | 0 | 230 | 2.7 | 108 | 2.6 |
| **AFJL** | 0 | 0 | 1 | 121 | 2.7 | 108 | 2.6 |
| **AFJL** | 0 | 1 | 0 | **92** | 3.3 | 108 | 2.6 |
| **AFJL** | 1 | 0 | 0 | 345 | 3.0 | 191 | 3.5 |
| **AFJL** | 1 | 0 | 1 | 150 | 3.0 | 190 | 3.5 |
| **AFJL** | 1 | 1 | 0 | 111 | *3.8* | 191 | 3.5 |
| **QK** | 0 | 0 | 1 | 118 | **1.6** | **82** | **1.3** |
| **QK** | 1 | 0 | 1 | 146 | 1.9 | 161 | 1.7 |

## A.6 Search for the intrinsic dimension

In Table 10 we report the values of $D^*$ across the 3 seeds used in each search experiment. We see good agreement within a factor of 2. For SNLI the target accuracy to exceed was $\tau_{90} = 80.1\%$; for XSUM the value of Rouge1 to exceed was 36.6 while that of Rouge2 was 15.69. In the case of Rouge2 compressing BART in half would lead to a search with $D = 2^{26}$; in that case the final values of Rouge2 did not exceed 14.4, so it stayed well-below the required threshold that defines the intrinsic dimension $D_{int}$ using a 90% target value of the metric obtained by fine-tuning the full model.

Table 10: Values of $D^*$ returned by the search the intrinsic dimension $D_{int}$ using 3 different seeds. This shows the stability of our algorithm which doubles the dimension of the fine-tuning subspace after some compute budget if the target metric has not improved enough.

| TASK | METRIC | PROJECTION ALGORITHM | $D^*$ |
|------|--------|----------------------|-------|
| SNLI | ACCURACY | **FFD** | $2^{\{14,13,13\}}$ |
| SNLI | ACCURACY | **AFJL** | $2^{\{14,13,13\}}$ |
| SNLI | ACCURACY | **QK** | $2^{\{13,13,13\}}$ |
| XSUM | ROUGE1 | **FFD** | $2^{\{26,25,25\}}$ |
| XSUM | ROUGE1 | **AFJL** | $2^{\{24,24,25\}}$ |
| XSUM | ROUGE1 | **QK** | $2^{\{26,25,26\}}$ |

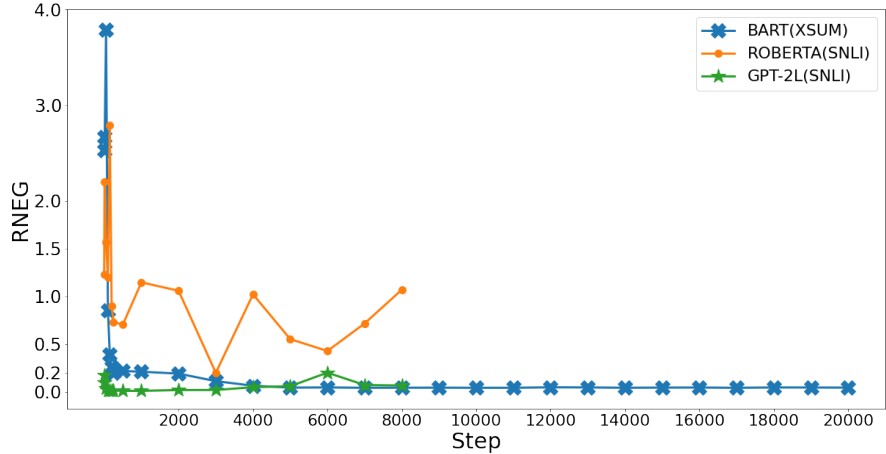

Figure 5: ratio (RNEG) of the absolute value of the top negative to the top positive eigenvalue

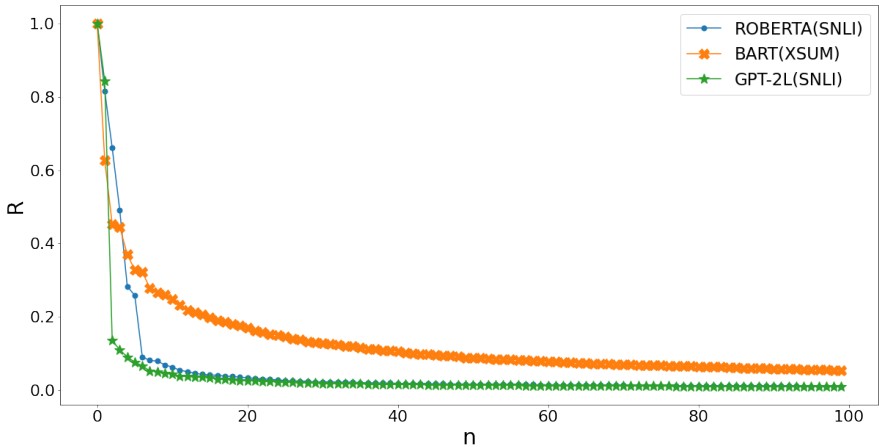

Figure 6: ratio $R$ of the $n$-th largest positive eigenvalue to the largest positive eigenvalue. We define outliers when $R > 20\%$, motivated by [12, Fig.2]

### A.7 Higher resolution version of Fig. 2

Figs. 5 and 6 are high-resolution versions of Fig. 2.

### A.8 Comparison to on-the-fly dense random projections

[19] proposes to materialize the full random projection on-the-fly in chunks (we will refer to this sketching method as TRAK, even though the full TRAK attribution method in [19] is more involved). This leads to two significant drawbacks: (1) the run-time scales linearly with the target dimension (memory traded off with compute), and (2) specialized kernels are necessary for efficient implementation, with unclear applicability to TPUs. Our attempts to implement a competitive version using pure JAX were unsuccessful due to the lack of control over memory allocation and placement. We have included a plot (Figure 7) and a table (Table 11) demonstrating the linear runtime growth of TRAK compared to the constant runtimes of **AFFD** and **QK**. The implementation challenges of TRAK are further highlighted by the fact that our Triton implementation does not outperform the original CUDA

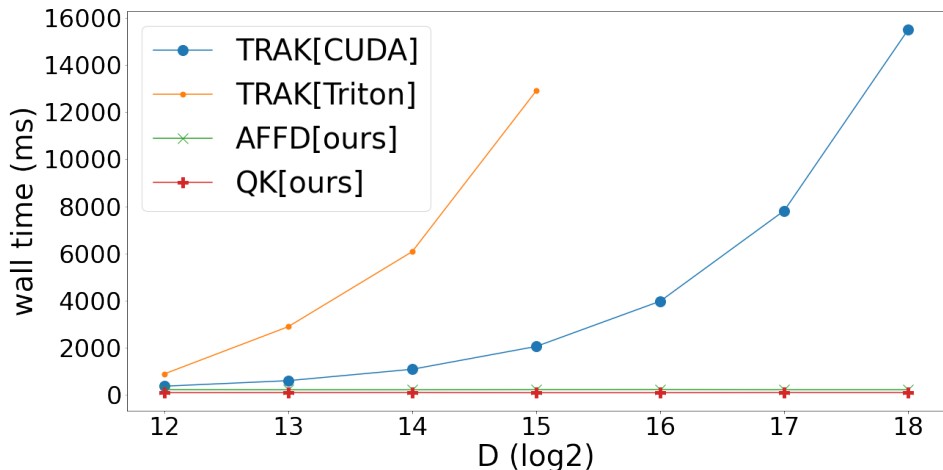

Figure 7: Our methods exhibit constant wall time with respect to the target dimension $D$. In contrast, TRAK's runtime increases with the target dimension. Efficient implementation of dense random projections with recomputed projectors is non-trivial; compare the performance difference between TRAK[CUDA] and TRAK[Triton]. TRAK[CUDA] utilizes the CUDA kernel provided by the original TRAK authors [19].

Table 11: Wall-time $T$ (ms) Comparison: Our Methods vs. On-the-fly Dense Projections (TRAK) using a V100 GPU. TRAK requires custom kernels and is thus restricted to GPU computation. Our methods exhibit constant runtime with respect to the target dimension, whereas TRAK's runtime increases substantially as the target dimension grows.

| $\log_2(D)$ | TRAK[CUDA] | TRAK[TRITON] | **AFFD**[OURS] | **QK**[OURS] |
|---|---|---|---|---|
| 12 | 359 | 879 | 213 | 84 |
| 13 | 594 | 2885 | 207 | 87 |
| 14 | 1078 | 6073 | 207 | 91 |
| 15 | 2047 | 12900 | 215 | 82 |
| 16 | 3969 | - | 219 | 82 |
| 17 | 7812 | - | 210 | 91 |
| 18 | 15500 | - | 209 | 90 |

kernel released by the TRAK authors, underscoring the difficulty of efficiently implementing random projections in chunks.

# B   Appendix: Implementation details

## B.1   Libraries, Compute resources and implementation of FFD and FJL

We use Jax and Hugging Face libraries; experiments in Sec. 5.2 were carried out using one GPU V100 or a TPUv2 (8 cores). Experiments in Sec. 5.4 used 2 V100s in the classification setting and 2 A100s in the generation setting. Experiments in Sec. 5.5 used 2 A100s. Experiments were performed on cloud infrastructure and the virtual machines employed for each experiment had at most 32GB of RAM. We used the Jax profiler tool to extract information about peak memory usage and wall time. We checked that under multiple runs the wall time estimates reported by the profiler tool stay within a 10% relative error, while the peak memory usage does not significantly change. For **FJL** and **FFD** the lookup operation is implemented as in previous work [1] storing the permutations or entries to sample using a vector. Regarding permutations there is an important implementation detail: if one wants the implementation of **FFD** to give the same results in implicit and explicit mode the

permutation used needs to be inverted between the two setups. For **FJL**, in implicit mode the lookup operation needs to be transposed and is implemented as an XLA scatter-add.

## B.2 HVP in Implicit vs Explicit Form

The implicit implementation of the HVP is easy; one changes the loss to be defined on the image of the projection $\Phi$ as follows:

$$L_{implicit}(\omega) = L(\theta_0 + \Phi^T \omega), \tag{10}$$

and then one computes the HVP at the origin $\omega = 0$. For the explicit form of the HVP, one takes a vector $\nu$ in the image of $\Phi$ and lifts it to the vector $\Phi^T \nu$ in the parameter space; then one computes the standard HVP for $\Phi^T \nu$ and applies $\Phi$ to the result obtaining again a vector in the image of $\Phi$.

## B.3 Step-by-step didactic implementation

We describe a step-by-step didactic implementation in Jax and Flax.

First, we rely on the following modules.

```
import jax
import jax.numpy as jnp
from typing import Sequence, Tuple
from flax.core import apply
from flax.core import init
from flax.core import nn
import scipy
import functools
```

The core subroutine is applying pre-conditioners using Kronecker products. One can use the Jax' einsum operation to write it in a few lines of code.

Listing 1: Kronecker Implemetation

```
def compute_kronecker_shapes(*, dimension: int) -> Tuple[int,
    ...]:
  """Computes shapes to decompose Kronecker products."""

  # For realistic use cases, bump it up, e.g. 1024
  max_block_size = 32

  n = dimension

  # Divide n into block sizes
  shape = []
  while n > 1:
    shape.append(min(n, max_block_size))
    n //= max_block_size
  shape.reverse()
  return tuple(shape)

def kronecker_product(
    *, x: jnp.ndarray, matrices: Sequence[jnp.ndarray]
) -> jnp.ndarray:
  """Performs kronecker product using Jax einsum."""

  shape = tuple(map(lambda x: x.shape[0], matrices))

  y = x.reshape(shape)

  num_dims = len(shape)
  # Einsum iterative implementation.
  for i, m in enumerate(matrices):
    y_dims = ''.join(str(j) for j in range(num_dims))
    h_dims = f'{i}{num_dims + 1}'
```

```
      out_dims = y_dims.replace(str(i), str(num_dims + 1), 1)
      operands = f'{y_dims},{h_dims}−>{out_dims}'
      y = jnp.einsum(operands, y, m)
   return y.flatten()
```

The next step is implementation of **QK** which is quite straightforward.

Listing 2: **QK** Implemetation

```
def q_init(rng, shape, dtype=jnp.float32):
   """Random orthogonal matrix initializer."""

   x = jax.random.normal(rng, shape=shape)
   q, _ = jnp.linalg.qr(x, mode='complete')
   return q.astype(dtype)

# target_dim = D in paper
# input_dim = N in paper
# vec is the N−vector to sketch to a D−vector.
def qk(scope, vec, target_dim, input_dim):
   """Implements QK."""

   shapes = compute_kronecker_shapes(dimension=input_dim)
   sigma = jnp.sqrt(input_dim/target_dim)
   params = []
   for i, s in enumerate(shapes):
     p = scope.param(f'q_{i}', q_init, shape=(s, s))
     params.append(p)
   return sigma * kronecker_product(x=vec, matrices=params)[:
       target_dim]
```

For using **QK** in implicit mode we need to transpose **QK**; we illustrate how this can be carried out in Flax:

Listing 3: Transpose of **QK**

```
# The tranpose of QK.
# vec is a D−vector to lift to an N−vector.
def qk_transpose(scope, vec, target_dim, input_dim):

   shapes_1 = compute_kronecker_shapes(dimension=input_dim)
   shapes_2 = compute_kronecker_shapes(dimension=target_dim)
   sigma = jnp.sqrt(input_dim/target_dim)

   params = []
   for i, (s_1, s_2) in enumerate(zip(shapes_1, shapes_2)):
     p = scope.param(f'q_{i}', q_init, shape=(s_1, s_1))
     params.append(p.T[:s_2])
   return sigma * kronecker_product(x=vec, matrices=params)
```

The implementation of other algorithms is more challenging; we include an implementation of **AFFD**; it is easy to figure out the implementations of the others (note that for **FFD** one needs to start with the implementation of the transpose because **FFD** is defined as a random feature generator).

Listing 4: Implementation of **AFFD**

```
def init_hadamard(rng, shape: Tuple[int, int], permute_col: bool)
    −> Sequence[jnp.ndarray]:
   """Creates randomly permuted hadamard matrices."""

   matrix = jnp.array(scipy.linalg.hadamard(shape[0]))
   pi = jax.random.permutation(rng, shape[0])
   if permute_col:
     matrix = matrix.at[:, pi].get()
   else: # permute rows
```

```
      matrix = matrix.at[pi, :].get()

  return matrix

# target_dim = D in paper
# input_dim = N in paper
# vec is the N-vector to sketch to a D-vector.
def affd(scope, vec, target_dim, input_dim):
  """Implements AFFD."""

  shapes = compute_kronecker_shapes(dimension=input_dim)
  sigma = jnp.sqrt(input_dim/target_dim)

  def init_ber(rng, shape, dtype=jnp.int32):
    return jax.random.choice(rng, jnp.array([-1, 1], dtype='int32'
      ),
                                 shape)

  b = scope.param('B', init_ber, (input_dim,))
  vec = vec * b
  h_1_params = []
  for i, s in enumerate(shapes):
    h_1 = scope.param(f'H_1_{i}', init_hadamard, shape=(s, s),
                      permute_col=False)
    h_1_params.append(h_1)

  vec = kronecker_product(x=vec, matrices=h_1_params) / jnp.sqrt(
    input_dim)

  def init_gauss(rng, shape, dtype=jnp.float32):
    return jax.random.normal(rng, shape=shape, dtype=dtype)

  g = scope.param('G', init_gauss, (input_dim,))
  vec = vec * g

  h_2_params = []
  for i, s in enumerate(shapes):
    h_2 = scope.param(f'H_2_{i}', init_hadamard, shape=(s, s),
                      permute_col=True)
    h_2_params.append(h_2)

  vec = kronecker_product(x=vec, matrices=h_2_params) / jnp.sqrt(
    input_dim)

  vec = sigma * vec[:target_dim]

  return vec
```

To transpose **AFFD**, we just need to reverse the above steps and transpose application of the Hadamard matrices.

Listing 5: Transpose of **AFFD**

```
# The tranpose of AFFD.
# vec is a D-vector to lift to an N-vector.
def affd_transpose(scope, vec, target_dim, input_dim):

  shapes_1 = compute_kronecker_shapes(dimension=input_dim)
  shapes_2 = compute_kronecker_shapes(dimension=target_dim)
  sigma = jnp.sqrt(input_dim/target_dim)

  h_2_params = []
  for i, (s_1, s_2) in enumerate(zip(shapes_1, shapes_2)):
    h_2 = scope.param(f'H_2_{i}', init_hadamard, shape=(s_1, s_1),
```

```
                          permute_col=True)
  h_2_params.append(h_2.T[:s_2])

vec = kronecker_product(x=vec, matrices=h_2_params) / jnp.sqrt(
    input_dim)

def init_gauss(rng, shape, dtype=jnp.float32):
  return jax.random.normal(rng, shape=shape, dtype=dtype)

g = scope.param('G', init_gauss, (input_dim,))
vec = vec * g

h_1_params = []
for i, s in enumerate(shapes_1):
  h_1 = scope.param(f'H_1_{i}', init_hadamard, shape=(s, s),
                    permute_col=False)
  h_1_params.append(h_1.T)

vec = kronecker_product(x=vec, matrices=h_1_params) / jnp.sqrt(
    input_dim)

def init_ber(rng, shape, dtype=jnp.int32):
  return jax.random.choice(rng, jnp.array([-1, 1], dtype='int32'
    ),
                           shape)

b = scope.param('B', init_ber, (input_dim,))
vec = vec * b

vec = vec * sigma

return vec
```

We now turn to a didactic implement of sketching gradients of a loss functions in implicit and explicit mode. We first make an assumption about the signature of loss and sketching functions

```
def loss_fn(model_params, batch):
  """Loss function signature."""
  pass

def sketch_fn(sketch_params, vec):
  """Sketch function signature."""
  pass

def transpose_sketch_fn(sketch_params, vec):
  """Transpose of sketch_fn signature."""
  pass
```

Then here's how one can implement explicit and implicit gradient sketching in a few lines of code.

Listing 6: Implementation of Implicit and Explicit Gradient Sketching

```
def explicit_grad_sketch(model_params, sketch_params, batch):
  """Performs an explicit gradient sketch."""

  grad = jax.grad(loss_fn)(model_params, batch)
  return sketch_fn(sketch_params, grad)

def implicit_grad_sketch(model_params, sketch_params, batch,
    target_dim):
  """Performs an implicit gradient sketch."""

  def inner_loss_fn(omega):
    omega = transposed_sketch_fn(sketch_params, omega)
    return loss_fn(model_params + omega, batch)
```

```
  omega = jnp.zeros((target_dim,))
  grad = jax.grad(inner_loss_fn)(omega)

  return grad
```

Here's how one can do the same for the sketched HVP.

Listing 7: Implementation of Implicit and Explicit HVP Sketching

```
# Note tangent_params is a D-dimensional vector
def explicit_hvp_sketch(model_params, tangent_params,
    sketch_params, batch):
  """Performs an explicit HVP sketch."""

  tangent_params = transposed_sketch_fn(sketch_params,
      tangent_params)
  loss_ = functools.partial(loss_fn, batch=batch)
  grad_fn = jax.grad(loss_)
  hvp = jax.jvp(grad_fn, (model_params,), (tangent_params,))[1]
  return sketch_fn(sketch_params, hvp)

# Note tangent_params is a D-dimensional vector
def implicit_hvp_sketch(model_params, tangent_params,
    sketch_params, batch, target_dim):
  """Performs an implicit HVP sketch."""

  def inner_loss_fn(omega):
    omega = transposed_sketch_fn(sketch_params, omega)
    return loss_fn(model_params + omega, batch)

  omega = jnp.zeros((target_dim,))

  loss_ = functools.partial(inner_loss_fn)
  grad_fn = jax.grad(loss_)
  hvp = jax.jvp(grad_fn, (omega,), (tangent_params,))[1]
  return hvp
```

## B.4 Sketching and model parallelism

We take the case of **AFFD**, and show how the single device code may be lifted to code employing model parallelism.

First, we rely on the additional modules.

```
from jax.sharding import NamedSharding
from jax.experimental import shard_map
from jax.sharding import Mesh
from jax.sharding import PartitionSpec as P
from jax.sharding import NamedSharding as NS
from jax import lax
from jax import tree_util as tu
import numpy as np
```

We then define the device mesh; we assume 8 cores with 4-way model parallelism and 2-way data parallelism.

```
mesh = Mesh(
  np.array(
    jax.devices()).reshape(2,4), ('data', 'model',))
```

We now lift initialization and application of the FlaX modules to model-parallel code:

## Listing 8: Lift **AFFD** to model parallel code

```python
# target_dim = D in paper
# input_dim = N in paper
# vec is the N-vector to sketch to a D-vector.

def part_fn(pytree):
  """Partitions each parameter on the last dimension."""

  def inner_part_fn(p):
    out = (None,) * (p.ndim-1) + ('model',)
    return P(*out)
  return tu.tree_map(inner_part_fn, pytree)

def init_affd_mp(scope, vec, target_dim, input_dim):
  """Initializes AFFD for model-parallel code."""

  # bind dimensional arguments to make jax tracer happy with
  # jax.eval_shape.
  affd_init_fn = functools.partial(
    init(affd), target_dim=target_dim,
    input_dim=input_dim)

  _, params_shape = jax.eval_shape(affd_init_fn, rng, vec)
  params_part = part_fn(params_shape)

  # We need to redefine the input_dim because the code
  # is now executed on each model partition.
  affd_init_fn = functools.partial(
    init(affd), target_dim=target_dim,
    input_dim=input_dim // mesh.shape['model'])

  def init_fn(rng, vec):
    # different rng on each model slice
    rng = jax.random.fold_in(rng, lax.axis_index('model'))
    out, params = affd_init_fn(rng, vec)
    # The vector output is fully replicated and we need to sum
    # on the 'model' partitions
    out = lax.psum(out, axis_name='model')
    return out, params

  return shard_map.shard_map(
    init_fn,
    mesh=mesh,
    in_specs=(P(None,), part_fn(vec)),
    out_specs=(P(None,), params_part),
    )(rng, vec)

def apply_affd_mp(params, vec, target_dim, input_dim):
  """Applies AFFD for model-parallel code."""

  # We need to redefine the input_dim because the code
  # is now executed on each model partition.
  affd_apply_fn = functools.partial(
    apply(affd), target_dim=target_dim,
    input_dim=input_dim // mesh.shape['model'])

  def apply_fn(params, vec):
    out = affd_apply_fn(params, vec)
    # The vector output is fully replicated and we need to sum
    # on the 'model' partitions
    out = lax.psum(out, axis_name='model')
    return out
```

```
    return shard_map.shard_map(
      apply_fn,
      mesh=mesh,
      in_specs=part_fn((params, vec)),
      out_specs=P(None,),
      )(params, vec)
```

We now illustrate how to use the previous code.

```
# We create the vector x on a single device and partition
# it on the model axis.
rng = jax.random.PRNGKey(0)
target_dim = 128
input_dim = 1024
x = jax.random.normal(jax.random.fold_in(rng, 1), shape=(input_dim
    ,))
x_mp = jax.device_put(x, NS(mesh, P('model',)))

affd_x_mp, affd_params_mp = mesh(init_affd_mp)(
    jax.random.fold_in(rng, 3), x_mp, target_dim, input_dim)

# Consistency check
affd_x_mp_2 = mesh(apply_affd_mp)(
    affd_params_mp, x_mp, target_dim, input_dim)
assert affd_x_mp_2.shape == affd_x_mp.shape
jnp.linalg.norm(affd_x_mp_2 - affd_x_mp)
```

### B.5 Algorithm for searching the intrinsic dimension

Our algorithm for searching the intrinsic dimension is in Listing 9. Without loss of generality we assume that the target metric needs to be maximized, e.g. for the loss one might use the negative loss.

Listing 9: An algorithm that searches the intrinsic dimension

```
def finetune(model_params, D_max: int, d: int, c: int):
  """Finetune function signature.

  Finetunes for c steps in D_max dimensional subspace but
  zeros out the last D_max - d components of the gradient.
  Returns the updated model_params.
  """
  pass

def evaluate(model_params):
  """Eval function signature."""
  pass

def search_intrinsic_dimension(
    model_params, D_min: int, D_max: int,tau_target: float,
    c: int, delta: float):
  """
  model_params: initial model parameters.
  D_min: start value for the intrinsic dimension.
  D_max: maximum allowed value of the intrinsic dimension.
  tau_target: desired target metric.
  c: number of finetuning steps in which we expect improvement.
  delta: minimum expected improvement.
  """

  d = D_min
  tau_old = evaluate(model_params)
  while True:
    model_params = finetune(model_params, D_max, d, c)
```

```
tau_new = evaluate(model_params)
if tau_new >= tau_target:
    return d
else if tau_new − tau_old < delta:
    d = 2 * d
    if d > D_max: raise ValueError("D_max exceeded")
tau_old = tau_new
```

### B.6 Hyper-parameters for Sections 5.2 and 5.3

Models were trained with the released code from [6]; then checkpoints were converted to Jax for benchmarking the sketching algorithms.

### B.7 Hyper-parameters for Sec. 5.4

Roberta was fine-tuned with a batch size of 32 for 10k steps with Adam and a constant learning rate of $2 \times 10^{-5}$. For the search algorithm 9 the learning rate was increased to $10^{-4}$, $\delta = 0.1$ and $c = 2k$ steps.

BART was fine-tuned with a batch size of 32 for 20k steps with Adam and a constant learning rate of $2 \times 10^{-5}$. For the search algorithm 9 the learning rate was increased to $10^{-4}$; $\delta_{Rouge1} = 0.5$, $\delta_{Rouge2} = 0.5$ and $c = 2k$ steps; the total number of steps was increased to 40k.

### B.8 Hyper-parameters for Sec. 5.5

We consider the SGD optimizer as in previous work [12]; the batch size was 32, the learning rate set to $10^{-5}$.

## C Appendix: Theory

### C.1 Definition of Higher order sketches

For higher-order derivatives of $L$, one can consider sketches of operators. For example the Hessian vector product is the operator $\mathrm{HVP} : \mathbb{R}^N \to \mathbb{R}^N$ given by $\mathrm{HVP}(u) = \nabla^2 L(\theta)(u)$, i.e. $\mathrm{HVP}(u)_i = \sum_j \partial^2_{i,j} L(\theta) u_j$. A sketch of the Hessian vector product can then obtained as follows: $\mathcal{S}(O)(v) = \Phi(\mathrm{HVP}(\Phi^T v))$ where $v \in \mathbb{R}^D$ so that we obtain an operator mapping $\mathbb{R}^D \to \mathbb{R}^D$. Sketches of matrices were extensively studied [22] to speed-up evaluation of matrix products. Extending the HVP-example, for an operator $O$ mapping $\mathbb{R}^{kN}$ to $\mathbb{R}^{sN}$ the transpose $\Phi^T$ is applied to the $k$ input indices and $\Phi$ is applied to the output $s$ indices to obtain a mapping $\mathcal{S}(O) : \mathbb{R}^{kD} \to \mathbb{R}^{sD}$ via:

$$(\mathcal{S}(O)v)_{l_1 \cdots l_s} = \sum_{t_1 \cdots t_s = 1}^{N} \sum_{i_1 \cdots i_k = 1}^{N} \sum_{j_1 \cdots j_k = 1}^{D} \prod_{\beta=1}^{s} \Phi_{l_\beta, t_\beta} \cdot O_{i_1 \cdots i_k; t_1 \cdots t_s} \cdot \prod_{\alpha=1}^{k} \Phi_{j_\alpha, i_\alpha} \cdot v_{j_1 \cdots j_k}. \quad (11)$$

### C.2 Guarantees on distorting distances.

By the method of Johnson-Lindenstrauss [10] one can leverage the equation about concentration of the sketched norm (1) to prove that, given $M$ points in $\mathbb{R}^N$, the distances between their sketches in $\mathbb{R}^D$ are distorted by at most a multiplicative factor $1 \pm \varepsilon$. The point is that concentration arguments establish, for the $\delta$ appearing in (1), a bound of the form $\delta = O(\exp(-\varepsilon^2 \beta^2))$: one can thus apply (1) to the $\frac{M(M-1)}{2}$ differences between points by requiring that $\frac{\beta}{\sqrt{\log M}}$ is sufficiently large; this is a considerable gain replacing a bound in terms of $M^2$ with one involving $\log M$.

### C.3 Definition of the Walsh-Hadamard transform.

The Fastfood paper [13] defines $H_N$ without scaling, so it is not an isometry; however we follow the definition with scaling as in Wikipedia so that $H_N$ is an isometry. Specifically, $N$ needs to be a power of 2; then for $i \leq N$ let $\Delta(i)$ denote the vector of 0s and 1s and of length $\log_2 N$, representing $i$ in its binary form; then $(H_N)_{i,j} = \frac{1}{\sqrt{N}}(-1)^{\langle \Delta(i), \Delta(j) \rangle}$, where $\langle, \rangle$ denotes the inner product.

### C.4 Failure of concentration for FFD.

**Theorem C.1.** *There are some inputs $x$ such that **FFD**$(x)$ does not satisfy* (1).

*Proof.* Intuitively, the problem with **FFD** is that transposition fails to apply the pre-conditioner to some bad inputs. Recall that the **FFD**, when used for sketching, gets transposed because it is applied in implicit form, i.e. generating random features that perturb the model parameters. The transposed operation of concatenation gives rise to a sum; specifically, given a unit vector $x \in \mathbb{R}^N$, we decompose it into $N/D$-blocks of size $D$, denoting the $b$-th block by $x_b$. Then

$$\mathbf{FFD}(x) = \sigma \sum_{b=1}^{N/D} B_b H \Pi_b G_b H(x_b); \tag{12}$$

then choose $x$ such that all blocks $x_b = 0$ for $b > 1$ and $x_1$ is such that $H(x_1) = e_1$, the first coordinate vector in $\mathbb{R}^D$. Then

$$\|\mathbf{FFD}(x)\|_2^2 = \sigma^2 g_1^2, \tag{13}$$

where $g_1$ is the first entry of $G_v$; then we must have $\sigma^2 = 1$ and $\|\mathbf{FFD}(x)\|_2$ cannot concentrate around 1 because $g_1$ has unit variance. $\qquad\square$

### C.5 Comparison to Kronecker products in [17]

[17] proposes two projection operators. The first one is

$$P_\oplus = \sigma \cdot (I \otimes R_1 + R_2 \otimes I) \tag{14}$$

where $I$ is a vector of ones in $\mathbb{R}^{\sqrt{N}}$ and $R_i$ is Gaussian of shape $D \times \sqrt{N}$ so that the memory cost of $P_\oplus$ is $O(D\sqrt{N})$. The second one is

$$P_\otimes = \sigma \cdot Q_1 \otimes Q_2, \tag{15}$$

where $Q_i$ is Gaussian of shape $\sqrt{D} \times \sqrt{N}$ so that the memory cost of $P_\otimes$ is $O(\sqrt{D}\sqrt{N})$. Our **QK** proposal is more general as it calls for a more general Kronecker decomposition

$$Q = \sigma \cdot Q^{(1)} \otimes Q^{(2)} \otimes \cdots \otimes Q^{(K)}; \tag{16}$$

where $Q^{(i)}$ is of shape $D_i \times B_i$, $\prod_i D_i = D$ (reconstruction of the target dimension) and $\prod_i B_i = N$ (reconstruction of the model dimension). So if we choose $K = 2$, $D_i = \sqrt{D}$ and $B_i = \sqrt{N}$ we recover $P_\otimes$. Strictly speaking, $P_\oplus$ is different from $P_\otimes$, but one might reconstruct it by averaging two of our $Q$'s (16), both defined with with $K = 2$: indeed, we select $Q_1$ where $Q^{(1)}$ is the vector $I/\sqrt{N}$ as in (14) and $Q^{(2)}$ is of shape $D \times \sqrt{N}$; we then select $Q_2$ where $Q^{(2)}$ is $I/\sqrt{N}$ and $Q^{(1)}$ is of shape $D \times \sqrt{N}$; in other words, to get $P_\oplus$ we restrict the sampling of one factor to $I/\sqrt{N}$. Note that memory-wise our approach is more efficient than [17] as we allow $K > 2$; memory cost is $O(\sum_{i=1}^K B_i D_i)$ which, choosing $B_i \simeq A$ and $D_i \simeq A'$ allows for a memory cost $O(AA' \log N)$. An implementation difference with [17] is that we sample from the special orthogonal group: we sample a Gaussian of shape $D_i \times B_i$ and obtain $Q^{(i)}$ using the QR-decomposition.

### C.6 Concentration result for QK: Theorem 3.3

**Theorem C.2.** *Consider the projection algorithm **QK** where $Q$ decomposes as $Q^{(1)} \otimes \cdots \otimes Q^{(K)}$ where $Q^{(i)}$ has shape $D_i \times B_i$; then*

$$P\Big(\sqrt{\frac{D}{N}}(1-\varepsilon)\|x\|_2 \leq \|Q(x)\|_2 \leq \sqrt{\frac{D}{N}}(1+\varepsilon)\|x\|_2\Big) \geq 1 - 2\sum_i \exp(-4CD_i((1+\varepsilon)^{1/K}-1)^2). \tag{17}$$

In particular, as long as each $D_i$ is sufficiently large one still obtains a concentration result; the price to pay for the Kronecker product decomposition is that the concentration probability is dampened by the number of factors as $((1+\varepsilon)^{1/K} - 1)^2 \simeq (\frac{\varepsilon}{K})^2$.

*Proof.* **Step 1: Reduction to the case of applying a single factor in the Kronecker product representation of $Q$.** We assume that $Q$ decomposes as $Q^{(1)} \otimes \cdots \otimes Q^{(K)}$ where $Q^{(i)}$ has shape $D_i \times B_i$; if we reshape $x$ into a tensor of shape $(B_1, \cdots, B_K)$ indexed by $(a_1, \cdots, a_K)$, the output $Q(x)$ can be represented as a tensor of shape $(D_1, \cdots, D_K)$ indexed by $(b_1, \cdots, b_K)$:

$$Q(x)_{b_1, \cdots, b_K} = \sum_{a_1, \cdots, a_K} Q^{(1)}_{b_1, a_1} Q^{(2)}_{b_2, a_2} \cdots Q^{(K)}_{b_K, a_K} x_{a_1, \cdots a_K}. \tag{18}$$

We now look into applying (18) one step at a time. We reshape $x$ to shape $(B_1, B_2 \cdots B_K)$ obtaining a matrix $X^{(1)}_{a_1, c}$; we then contract it with $Q^{(1)}_{b_1, a_1}$ over $a_1$ to obtain a matrix $Y^{(1)}_{b_1, c}$ of shape $(D_1, B_2 \cdots B_K)$. To apply $Q^{(2)}_{b_2, a_2}$ we need to first reshape $Y^{(1)}$ to shape $(D_1, B_2, \cdots, B_K)$, then transpose the first and second indices, and reshape it to a matrix $X^{(2)}_{a_2, c}$ of shape $(B_2, D_1 B_3 \cdots B_K)$; we can then contract it with $Q^{(2)}_{b_2, a_2}$ over $a_2$ to obtain a matrix $Y^{(2)}_{b_2, c}$ of shape $(D_2, D_1 B_3 \cdots B_K)$. It should be clear how this procedure can be continued for each $i \in \{3, \cdots K\}$. Assume that for each $i$ we can prove that the Frobenius norm (which is the $l^2$-norm if we reshape it to be a vector) of $Y^{(i)}_{b_i, c}$ concentrates around that of $X^{(i)}_{b_i, c}$ up to a multiplicative factor $\sqrt{\frac{D_i}{B_i}}$:

$$P\Big(\sqrt{\frac{D_i}{B_i}}(1 - \varepsilon_i)\|X^{(i)}\|_2 \le \|Y^{(i)}\|_2 \le \sqrt{\frac{D_i}{B_i}}(1 + \varepsilon_i)\|X^{(i)}\|_2\Big) \ge 1 - \delta_i; \tag{19}$$

by conditional independence of the matrices $Q^{(i)}$ on one another we get that

$$P\Big(\sqrt{\frac{D}{N}}\prod_{i=1}^{K}(1 - \varepsilon_i)\|x\|_2 \le \|Q(x)\|_2 \le \sqrt{\frac{D}{N}}\prod_{i=1}^{K}(1 + \varepsilon_i)\|x\|_2\Big) \ge \prod_{i=1}^{K}(1 - \delta_i), \tag{20}$$

where we used $\prod_{i=1}^{K} D_i = D$ and $\prod_{i=1}^{K} B_i = N$.

**Step 2: Using concentration of measure on the orthogonal group.** The entries of $Q^{(i)}$ are not independent because of the orthogonality requirement and the fact that the rows need to have $l_2$-norm equal to 1. We will employ measure concentration without independence; as a reference for notation and theorems we use [26]. From [26, 2.5.2] we recall the definition of the sub-Gaussian norm of a real-valued random variable $X$ as:

$$\|X\|_{\psi_2} = \inf\{t > 0 : E \exp(X^2/t^2) \le 2\}; \tag{21}$$

obtaining a bound on $\|X\|_{\psi_2}$ is equivalent to a concentration inequality:

$$P(|X| \ge t) \le 2 \exp(-ct^2/\|X\|_{\psi_2}^2), \tag{22}$$

for a universal constant $c > 0$. Note that $Q^{(i)}$ can be sampled on the orthogonal group $SO(B_i)$ by restricting to the first $D_i$-rows in the case in which $D_i < B_i$ (by sampling from $O(B_i)$ and changing in case the sign of one of the last $B_i - D_i$ rows to ensure the determinant is 1), while the result we are proving is trivial if $D_i = B_i$ because then $Q^{(i)}$ is full-rank. We now invoke the concentration of measure for $SO(B_i)$ [26, 5.2.7]; if $f : SO(B_i) \to \mathbb{R}$ is Lipschitz:

$$\|f(Q^{(i)}) - Ef(Q^{(i)})\|_{\psi_2} \le C \frac{\|f\|_{Lip}}{\sqrt{B_i}}, \tag{23}$$

where $C$ is a universal constant and the Lipschitz constant $\|f\|_{Lip}$ is computed using the Frobenius norm on the tangent space. We now define:

$$f(Q^{(i)}) = \|Y^{(i)}\|_2 = \Big(\sum_{b_i, c}\Big(\sum_{a_i} Q^{(i)}_{b_i, a_i} X^{(i)}_{a_i, c}\Big)^2\Big)^{\frac{1}{2}}; \tag{24}$$

which has derivative:

$$\frac{\partial f(Q^{(i)})}{\partial Q_{b_i, a_i}} = \frac{\sum_c Y^{(i)}_{b_i, c} X^{(i)}_{a_i, c}}{\|Y^{(i)}\|_2}; \tag{25}$$

the Cauchy-Schwartz inequality implies that:

$$\left| \frac{\partial f(Q^{(i)})}{\partial Q_{b_i,a_i}} \right| \leq \frac{(\sum_c (Y_{b_i,c}^{(i)})^2)^{1/2} (\sum_c (X_{a_i,c}^{(i)})^2)^{1/2}}{\|Y^{(i)}\|_2}, \tag{26}$$

from which it follows that $\|f\|_{Lip} \leq 1$ if one assumes that $\|X^{(i)}\|_2 \leq 1$. Note that to derive (20) we may rescale $x$ by a constant; if we rescale it so that $\|x\|_2 = 1$ then all the norms of the intermediate results $\|X^{(i)}\|_2$, $\|Y^{(i)}\|_2$ are at most 1 because the matrices $Q^{(i)}$ are orthogonal. We have thus established

$$\|f(Q^{(i)}) - Ef(Q^{(i)})\|_{\psi_2} \leq \frac{C}{\sqrt{B_i}}. \tag{27}$$

**Step 3: Replacing $Ef(Q^{(i)})$ with something simpler to estimate.** A drawback of (27) is that the term $Ef(Q^{(i)})$ is not easy to estimate. However, using a symmetry argument, it is easy to estimate $E(f(Q^{(i)}))^2$; indeed the $D_i$ variables $\sum_c (Y_{b_i,c}^{(i)})^2$ are identically distributed and if $D_i = B_i$ one would get an isometry; so

$$E(f(Q^{(i)}))^2 = \frac{D_i}{B_i} \|X^{(i)}\|_2^2. \tag{28}$$

So we would like to replace $Ef(Q^{(i)})$ with $\sqrt{E(f(Q^{(i)}))^2}$; the intuition why this would work is that concentration around the mean is equivalent to concentration around the median; so $Ef(Q^{(i)})$ concentrates around the median $M_i$; as $f(Q^{(i)})$ is non-negative, the median of $(f(Q^{(i)}))^2$ is $M_i^2$; and this variable concentrates both around the mean and the median, and we have a closed form for the mean (28). To make this more precise, by the fact that concentration around the mean is equivalent to concentration around the mean (see [26, 5.1.13]), we have

$$\|f(Q^{(i)}) - M_i\|_{\psi_2} \leq \frac{C}{\sqrt{B_i}}, \tag{29}$$

where the constant $C$ might have changed but is universal. We then just need to show that $|\sqrt{E(f(Q^{(i)})^2)} - M_i|$ is $O(1/\sqrt{B_i})$. By the triangle inequality:

$$|\sqrt{E(f(Q^{(i)})^2)} - M_i| \leq \sqrt{E|f(Q^{(i)}) - M_i|^2}, \tag{30}$$

and we can compute the right hand side with the layer cake decomposition:

$$\sqrt{E|f(Q^{(i)}) - M_i|^2} = \left( \int_0^\infty P(|f(Q^{(i)}) - M_i|^2 \geq u) \, du \right)^{\frac{1}{2}}, \tag{31}$$

and we apply the concentration inequality (22) to the right hand side to get

$$\sqrt{E|f(Q^{(i)}) - M_i|^2} \leq \left( \int_0^\infty 2 \exp(-\tilde{c} B_i u) \, du \right)^{\frac{1}{2}}, \tag{32}$$

which implies that the right hand size is $O(1/\sqrt{B_i})$. We have thus established

$$P\left( \left| \|Y^{(i)}\|_2 - \sqrt{\frac{D_i}{B_i}} \|X^{(i)}\|_2 \right| \geq t \right) \leq 2 \exp(-C B_i t^2). \tag{33}$$

We now deduce (19) conditional that it holds for $j = 1, \cdots, i-1$ so that we may assume that $\|X^{(i-1)}\|_2 \geq \frac{1}{2}$; then we may take $t = \varepsilon_i \sqrt{\frac{D_i}{B_i}} \|X^{(i)}\|_2$ and get (19) with

$$\delta_i = 2 \exp(-4 C D_i \varepsilon_i^2). \tag{34}$$

**Step 4: Choosing the $\varepsilon_i$.** We just aim for $\varepsilon_i$ to be equal and that $\prod_i (1 + \varepsilon_i) = 1 + \varepsilon$; this is achieved by letting $\varepsilon_i = (1 + \varepsilon)^{1/K} - 1$. In this case we may lower bound $\prod_i (1 - \delta_i)$ by $1 - 2 \sum_i \exp(-4 C D_i ((1 + \varepsilon)^{1/K} - 1)^2)$. $\qquad \square$

## C.7 Concentration result for AFFD

**Theorem C.3.** *Consider sketching with **AFFD**; then*

$$P\Big((1-\varepsilon)\|x\|_2 \le \|\mathbf{AFFD}(x)\|_2 \le (1+\varepsilon)\|x\|_2\Big) \ge 1-\delta, \tag{35}$$

*where for each $\delta_1 > 0$ we have*

$$\delta \le \delta_1 + \exp\left(-C\varepsilon^2 \frac{N}{2\log\frac{2N}{\delta_1}}\right) + \exp\left(-C\varepsilon^2 \frac{D}{4\log^2\frac{2N}{\delta_1}}\right). \tag{36}$$

Note that the first two terms on the right hand side (36) can be made arbitrarily small (especially as $N$ is typically very large and $\delta_1$ will affect $C$ in the third term); thus the effective bound for $\delta$ is of the form $\delta \le \exp\left(-C\varepsilon^2 \frac{D}{\log^2 N}\right)$.

*Proof.* We will again use the notation and conventions from [26]. Let us recall the definition of **AFFD**:

$$\Phi(x) = R_D(\sigma \cdot H_2 \cdot G_v \cdot H_1 \cdot B(x)); \tag{37}$$

without loss of generality we will assume that $\|x\|_2 = 1$.

**Step 1: Using the pre-conditioner $H_1$ to distribute the mass of $x$.** Note that $H_1$ is an $N \times N$-dimensional matrix with entries of the form $\pm\frac{1}{\sqrt{N}}$; each entry of the vector $H_1 B(x)$ is of the form $\sum_i \frac{s_i b_i x_i}{\sqrt{N}}$ where the $\{b_i\}_{i=1}^N$ are independent Bernoulli and $s_i = \pm 1$; applying Hoeffding's inequality [26, Thm.2.2.2] to each entry of $H_1 B(x)$ we obtain that:

$$P\left(\|H_1 B(x)\|_\infty \ge \frac{t_\infty}{\sqrt{N}}\right) \le 2N \exp\left(-\frac{t_\infty^2}{2}\right); \tag{38}$$

intuitively, the norm of each entry of $H_1 B(x)$ cannot become much larger than a multiple of its variance $\frac{1}{\sqrt{N}}$: this is the purpose of using a pre-conditioner to distribute the mass of $x$.

**Step 2: Decomposing $\|\Phi(x)\|_2^2$.** We now let $u = H_1 \cdot B(x)$ so that $\Phi(x) = \sigma R_D(H_2 \cdot G_v \cdot u)$; we let $\Pi$ be the permutation associated with rearranging the columns of $H_2$ so that

$$\|\Phi(x)\|_2^2 = \sigma^2 \sum_{i=1}^D \left(\sum_{j=1}^N H_{i,\Pi(j)} g_j u_j\right)^2; \tag{39}$$

in (39) we decompose the effect of the diagonal and the off-diagonal terms obtaining

$$\|\Phi(x)\|_2^2 = \underbrace{\sigma^2 \sum_{i=1}^D \sum_{j=1}^N \frac{1}{N} g_j^2 u_j^2}_{T_1} + \underbrace{\sigma^2 \sum_{i=1}^D \sum_{k\neq j} H_{i,\Pi(k)} H_{i,\Pi(j)} g_j g_k u_j u_k}_{T_2}. \tag{40}$$

We now use the first term $T_1$ to compute $\sigma$:

$$ET_1 = \sigma^2 \sum_{j=1}^N \frac{D}{N} u_j^2, \tag{41}$$

where we used the fact the components of $G_v$ have unit variance; as $H_1 B(x)$ is an isometry, to have $ET_1 = 1$ we just need to set $\sigma = \sqrt{\frac{N}{D}}$.

**Step 3: Concentration for $\sqrt{T_1}$.** If we regard $\sqrt{T_1}$ as a function $f_1(G_v)$, we have

$$f_1(G_v) = \left(\sum_{j=1}^N g_j^2 u_j^2\right)^{1/2}; \tag{42}$$

conditional on the good event $E_{good}$ that $\|H_1 B(x)\|_\infty \le \frac{t_\infty}{\sqrt{N}}$, this function is $\frac{t_\infty}{\sqrt{N}}$-Lipschitz in the $l^2$-norm of $G_v$; applying concentration for the Gaussian measure on $\mathbb{R}^N$ [26, Sec. 5.2.2] we get that conditional on $E_{good}$

$$P\left(|\sqrt{T_1} - 1| \ge \varepsilon\right) \le \exp\left(-C\frac{N}{t_\infty^2}\varepsilon^2\right). \tag{43}$$

**Step 4: Concentration for $|T_2|$.** We would like to claim that $T_2$ is small with high probability. This is the second point in which we use the pre-conditioner $H$; the intuition is that applying the pre-conditioner $H$ to $G_v$ further reduces the finite sample correlation of the rows of the resulting matrix. Let us rewrite $T_2$ as follows:

$$T_2 = \sum_{k \ne j} \underbrace{\sum_{i=1}^{D} \frac{N}{D} H_{i,\Pi(k)} H_{i,\Pi(j)}}_{B_{ij}} u_j u_k g_j g_k; \tag{44}$$

so we have reduced $T_2$ to a bi-linear form $\sum_{j,k} B_{j,k} g_j g_k$; by the Hanson-Wright inequality [26, 6.2.1] we have

$$P(|T_2| \ge \varepsilon) \le \exp\left(-C \min\{\frac{\varepsilon^2}{\|B\|_F^2}, \frac{\varepsilon}{\|B\|_S}\}\right), \tag{45}$$

where $\|B\|_F$ is the Frobenious norm of $B$ and $\|B\|_S$ is the spectral norm. Let us look at $B_{j,k}$ conditional on $E_{good}$:

$$B_{j,k} = \sum_{i=1}^{D} \frac{N}{D} H_{i,\Pi(k)} H_{i,\Pi(j)} u_j u_k. \tag{46}$$

Recall now that $D$ and $N$ are both powers of 2 and the definition of $H$ in Sec. C.3: if $\Delta(j)$ denotes the binary vector, of length $\log_2 N$, representing an integer $j \le N$, we have $H_{i,\Pi(k)} = \frac{1}{\sqrt{N}}(-1)^{\langle \Delta(i), \Delta(\Pi(k))\rangle}$, where $\langle a, b\rangle$ denotes the inner product of two vectors of length $\log_2 N$. We thus obtain the bound:

$$|B_{j,k}| \le \frac{t_\infty^2}{D}\left|\frac{1}{N}\sum_{i=1}^{D}(-1)^{\langle \Delta(i), \Delta(\Pi(k))+\Delta(\Pi(j))\rangle}\right|; \tag{47}$$

the sum $\sum_{i=1}^{D}(-1)^{\langle \Delta(i), \Delta(\Pi(k))+\Delta(\Pi(j))\rangle}$ is 0 unless the vectors $\Delta(\Pi(j))$ and $\Delta(\Pi(k))$ agree in the first $\log_2 D$ entries, otherwise varying $i \le D$ we can always find two terms in the sum that cancel each other by flipping the parity of $i$ in the first slot where the vectors $\Delta(\Pi(j))$ and $\Delta(\Pi(k))$ differ. So for each $j$, there are at most $\frac{N}{D}$ possible $k$-s such that $|B_{j,k}|$ is non-zero; moreover, as the sum $\sum_{i=1}^{D}(-1)^{\langle \Delta(i), \Delta(\Pi(k))+\Delta(\Pi(j))\rangle}$ is at most $D$ in absolute value, we get

$$\|B\|_F^2 = \sum_{j=1}^{N}\sum_{k=1}^{N}|B_{j,k}|^2 \le N\frac{t_\infty^4}{D^2}\frac{N}{D}\frac{D^2}{N^2} \le \frac{t_\infty^4}{D}; \tag{48}$$

note that a bound on the spectral norm $\|B\|_S$ is trivial from (47) as we can just take the maximum of the absolute values of the $|B_{j,k}|$ which is bounded by $\frac{t_\infty^2}{N}$. Given the stronger bound for $\|B\|_S$ and that $\varepsilon \ll 1$, the minimum in the exponential in (45) is achieved by the term involving $\|B\|_F^2$ and we thus obtain:

$$P(|T_2| \ge \varepsilon) \le \exp\left(-C\varepsilon^2\frac{D}{t_\infty^4}\right). \tag{49}$$

**Step 5: Picking up constants.** Conditional on $E_{good}$ and on $\sqrt{T_1} \ge \frac{1}{2}$ we have:

$$|\sqrt{T_1 + T_2} - 1| \le |\sqrt{T_1} - 1| + |\sqrt{T_1 + T_2} - \sqrt{T_1}| \le |\sqrt{T_1} - 1| + 2|T_2|. \tag{50}$$

We can bound the right hand side of (50) by $3\varepsilon$ if $E_{good}$ and the concentration inequalities for $\sqrt{T_1}$ and $T_2$ hold. Thus, by decreasing the constant $C$ by a factor 9, we have

$$P\left(|\sqrt{T_1 + T_2} - 1| \ge \varepsilon\right) \le \delta, \tag{51}$$

where

$$\delta = 2N \exp\left(-\frac{t_\infty^2}{2}\right) + \exp\left(-C\frac{N}{t_\infty^2}\varepsilon^2\right) \exp\left(-C\varepsilon^2 \frac{D}{t_\infty^4}\right); \tag{52}$$

having fixed a small $\delta_1$, if we set $t_\infty = \sqrt{2\log\frac{2N}{\delta_1}}$, we get

$$\delta \le \delta_1 + \exp\left(-C\varepsilon^2 \frac{N}{2\log\frac{2N}{\delta_1}}\right) + \exp\left(-C\varepsilon^2 \frac{D}{4\log^2\frac{2N}{\delta_1}}\right). \tag{53}$$

$\square$

**Comparison with [4]**  It seems plausible that **Step 4** could be carried out with arguments similar to those of [4, Lem. 16, Lem. 17] by analyzing the chromatic number of the $P$-system associated with $H$; however we think the method that uses the Hanson-Wright inequality is more simple for this specific case.

