# OpenReview forum: "Efficient Sketches for Training Data Attribution and Studying the Loss Landscape"
_NeurIPS.cc/2024/Conference — NeurIPS 2024 poster_

### Official Review · Reviewer_xaAC · 2024-07-12

**Soundness:** 2
**Presentation:** 2
**Contribution:** 3
**Rating:** 6
**Confidence:** 4

**Summary:**

The paper proposes efficient sketching algorithms designed to address memory constraints in large-scale models. The key idea is to eliminate the multiplication of a dense projection matrix multiplication (large matrix materialization) found in the existing sketching algorithms such as in FJL and FastFood. The authors provide theoretical guarantees for their proposed algorithms: AFFD, AFJL, and QK. They showcase the advantages of these methods in three different contexts: training data attribution, intrinsic dimension estimation, and eigenvalue estimation of the Hessian.

**Strengths:**

- The paper addresses an important topic that is likely to be of interest to the NeurIPS community, with potential applications across various domains.
- The authors provide a well-motivated problem statement, clearly articulating the need for efficient sketching algorithms.
- The results presented are comprehensive and interesting, covering multiple aspects of the proposed algorithms and demonstrating their effectiveness across several use cases.
- While the authors have not provided code with the submission, the appendix contains sufficiently detailed information to enable the reproduction of the results.

**Weaknesses:**

- While the paper is generally well-written, there are several areas that require improvement: a) Some notations are not clearly explained. b) Certain statements lack precision, potentially leading to ambiguity (specific examples are provided in the detailed comments section).
- The authors do not sufficiently address the limitations of the proposed approach. The brief mention in the conclusion does not adequately address the potential drawbacks or constraints of the method (e.g., the limitations of QK).
- The paper would benefit from a more comprehensive comparative analysis, such as the sketching algorithm used in TDA [1, 2].

[1] Park, S. M., Georgiev, K., Ilyas, A., Leclerc, G., & Madry, A. (2023). Trak: Attributing model behavior at scale. arXiv preprint arXiv:2303.14186.

[2] Xia, M., Malladi, S., Gururangan, S., Arora, S., & Chen, D. (2024). Less: Selecting influential data for targeted instruction tuning. arXiv preprint arXiv:2402.04333.

**Questions:**

I have also included suggestions in addition to questions.

- The authors should be more specific when discussing Training Data Attribution (TDA) methods. Rather than using the broad term "TDA," they should explicitly mention that their work focuses on gradient-based TDA methods, such as TracIn [3] and GradDot [4]. TDA is a wide-ranging concept that encompasses various approaches, some of which do not align with the description provided in line 22.
- In the introduction, the authors state that random projections have been implemented using dense matrices for TDA. Several works perform random projections without explicitly materializing dense matrices. The same holds for the experiment section in 5.
- The statement “memory constraints limited their investigation of ID in generative tasks to 500k” in line 98 is unclear. What does ID stand for?
- In line 103, the authors claim that their work makes influence function computations more efficient. Please clarify this point, considering that influence functions typically require inverse Hessian-vector product (IHVP) estimation, which isn't directly addressed in Section 5.1. Is this claim primarily based on improving methods like the Arnoldi iteration?
- In Section 5.1, the conclusion that "layer selection coupled with dense projections faces severe scalability limitations" relative to AFFD, QK, and AFJL is not clear. Consider including tabular results (similar to Table 5) in the main paper to support this claim.
- Do you anticipate any challenges in applying these techniques to even larger language models?

Minor Comments

- Line 129: Replace "D" with "$D$" for consistent mathematical notation.
- Line 150: Clarify that $B$ and $H_D$ now have dimensions $D \times D$.
- Line 172: Specify the relevant section when mentioning "unacceptable early TPU results" for clearer cross-referencing.
- Line 227: Add a period at the end of the sentence.
- In the case of TDA, the rankings between the gradient dot products are important. Are there any reasons why authors do not use the Spearman correlation?
- In line 311, it would be helpful if the authors provided a direct explanation of why explicit sketching provides substation speed up.
- Figure 1 has low resolution, making it difficult to read when printed.

[3] Pruthi, G., Liu, F., Kale, S., & Sundararajan, M. (2020). Estimating training data influence by tracing gradient descent. Advances in Neural Information Processing Systems, 33, 19920-19930.

[4] Charpiat, G., Girard, N., Felardos, L., & Tarabalka, Y. (2019). Input similarity from the neural network perspective. Advances in Neural Information Processing Systems, 32.

**Limitations:**

The paper would benefit from a more comprehensive discussion of limitations.

---

> ### Author Rebuttal · Authors · 2024-08-06
>
> We sincerely thank the reviewer for their careful reading of our paper and the many valuable suggestions to improve the presentation. We also appreciate the reviewer's acknowledgment that this work is likely to be of interest to the NeurIPS community, with potential applications across various domains.
>
> **Answers to Questions**
>
> ```The authors should be more specific  ... description provided in line 22.```:  We acknowledge the need for greater specificity when discussing Training Data Attribution (TDA) methods. We commit to revise the manuscript to explicitly mention that our work focuses on gradient-based TDA methods, such as TracIn [3] and GradDot [4]. We recognize that TDA is a broad concept encompassing various approaches, and we commit to clarify that our work is not applicable to all TDA techniques. We commit also to clarify that we consider methods built on estimating the gradient (e.g., TracIn, GradDot) or quantities derived by preconditioning the gradient with the inverse Hessian ([10, Koh et al.], [21, Schioppa et al.]).
>
> ```In the introduction, the authors state that random projections ... experiment section in 5.```: We acknowledge that some works perform random projections without explicitly materializing dense matrices. However, the methods we are aware of (e.g., TRAK [
> https://arxiv.org/pdf/2303.14186, B.1]) still need to temporarily materialize chunks of the dense projection on-the-fly. This approach has two substantial disadvantages: (1) runtime grows linearly in the target dimension (they trade-off memory with compute), and (2) there is a need for specialized kernels for efficient implementation, and it's thus unclear how to implement this on TPUs. We have included a plot of runtimes (Figure 1) and a table (Table 1) in the rebuttal PDF to illustrate these points. We emphasize that implementation difficulty is a drawback of these methods, as evidenced by our Triton implementation of TRAK not outperforming the original CUDA kernel released by the TRAK authors. We also emphasize that in the case of using pure JAX we were not able to get a satisfactory implementation because one cannot control how the XLA compiler will deallocate these temporary arrays or their placement in the memory hierarchy. On the other hand, our methods can be written in pure JAX and have constant runtime in the
> target dimension.
>
> ```The statement ... What does ID stand for?```: *ID* stands for Intrinsic Dimension (line 90). We meant that [15, Liu et al.] could only work with a target sketching dimension <= 500k, which prevented them from searching for the true value of the intrinsic dimension (ID), as we demonstrate on a summarization task where ID approaches the model dimension.
>
> ```In line 103, ...  like the Arnoldi iteration?```:  Let us start with an observation from [7, Guo et al; Sec 3, eq(4)]: by symmetry of the Hessian, the iHVP can be either applied to the train or test points; however, given that the train data is much bigger, the iHVP should be applied just to the test point. Therefore the cost of an iHVP is limited only to test points. Now, our claim about improving influence function computations is based on two different approaches:
>
> 1. *Sketching the output of an iterative iHVP solver applied to the gradient.* This doesn't change the iHVP computation itself, but makes storage and the search for influential examples more efficient.
> 1. *Applying the Arnoldi iteration [21, Schioppa et al.] to the sketched Hessian, then using the sketched eigenvalues and eigenvectors to apply their method.* This is more efficient as it operates on vectors of dimension $D$ (sketching dimension) instead of the original model dimension $N$. Moreover, as we demonstrate in 5.4,
> we can scale the Arnoldi iteration well beyond to what was done in [11, Krishnan et al.] and [21, Schioppa et al.].
>
> In both cases storage is reduced from $N$ to $D$; in the second case the step of applying $k$ Arnoldi projectors can be performed very efficiently if $kD$ is sufficiently small to fit in the GPU/TPU memory as the computation can be parallelized across projectors.
>
> ```In Section 5.1 ... to support this claim.```: We commit to add this table.
>
> ```Do you anticipate ... larger language models?```: We focused on models that don't require partitioning weights across devices. For larger models, sketching can be applied to individual partitions. We commit to include an example in Appendix B demonstrating how to lift single-device code to multi-device code in JAX using shard-map to create sharded versions of sketching algorithms.
>
> ```In the case of TDA, ... Spearman correlation?```: We use Pearson correlation because we are comparing sketched or layer-selected dot products against full dot products, where a linear relationship is expected. Spearman correlation is more appropriate when the relationship is assumed monotonic but not necessarily linear.
>
> **Other**
>
> We commit to incorporate the suggested minor fixes and appreciate the reviewer pointing them out.  Regarding limitations, we are open to suggestions for expanding the discussion in Section 6 and would welcome specific areas the reviewer believes warrant further elaboration. At the moment we can think of emphasizing that QK might require higher values of the target dimension than AFFD.
>
> **Additional Clarifications Regarding LESS**
>
> We provide further clarification on the comparative analysis with LESS (TRAK is discussed above)
>
> LESS builds upon the sketching of TRAK by applying it on top of LoRA. However, as noted by reviewer kKVg, *full fine-tuning often outperforms LoRA fine-tuning in LLMs, a finding corroborated by our intrinsic dimension analysis*. Additionally, LESS focuses on data selection, which is orthogonal to the core insights of our work.

---

> > ### Comment · Reviewer_xaAC · 2024-08-11
> > **Reply**
> >
> > I thank the authors for their reply and acknowledge reading their response. As the authors mentioned, it would be helpful to be more explicit about the TDA in the motivation of the paper (and give a more detailed description of why this also applies to quantities derived by preconditioning the gradient), add explicit comparisons to Trak's random projection (as the author did for the general rebuttal and the response to my review), and address other details the authors mentioned they will fix/add in the next revision of the manuscript. In general, it seems that most reviewers, including myself, had trouble following the notations used in the paper, which made the presentation of the work on the weaker side. Relatedly, while minor, I agree with reviewer 9kYE that it was difficult to follow all the acronyms in the paper (e.g. AFFD, AFJL, and QK), and it would be helpful to fix issues with \citep vs \citet. It would make the paper more clear if the authors could address these issues. Given these issues will be resolved, as the authors promised, I have increased my score (but it was difficult for me to give a higher score since the current process does not allow the reviewers to see the final manuscript).

---

### Official Review · Reviewer_9kYE · 2024-07-12

**Soundness:** 3
**Presentation:** 2
**Contribution:** 2
**Rating:** 5
**Confidence:** 2

**Summary:**

The authors present a framework for scalable gradient sketching and HVP sketching. They introduce three algorithms AFFD, AFJL, QK and provide guarantees for sketching. The paper focuses on three applications: training data attribution, intrinsic dimension computation, and Hessian spectra analysis. These are all implemented with a focus on pre-trained language models.

**Strengths:**

* The scalable approaches to sketching are well-motivated and interesting. It seems important to figure out useful sketching routines that scale to high-dimensional parameter spaces that we are seeing in pre-trained language models.
* The theory appears to be a strength of the paper. It appears that there are two approaches taken in the paper, the direct approach that implements a gradient in the original parameter space and the implicit approach that implements a gradient in the sketched dimension. It is interesting to see the comparison made between these two different approaches.
* The introduction of the FFT as a faster pre-conditioner appears novel.

**Weaknesses:**

* Presentation is a weakness of the paper. Since the paper is proposing three different algorithms along with three different applications, it makes it challenging for a reader to follow the contributions and comparisons being made. For example, there seems to be two separate topics being discussed: 1) New theory/algorithms for scalable sketching, and 2) Implications for Pre-trained large language models. Additionally 2) aims to cover three different insights including layer selection, intrinsic dimension, and LLM Spectra. It is a challenge to cover all of the aforementioned points in enough detail in a 9 page paper.
* Table 1 is not clear how it is related to the proposed algorithms. In particular, TDA (section 5.1) just writes in bold “Our findings indicate the unreliability of layer selection (Table 1)”. Layer selection does not seem to be well defined, nor related to  AFFD, AFJL, QK.
* Table 3 is also unclear. Which algorithm should one use in practice?
* Minor: Figure 1 is hard to read and is too small. A single sentence in the caption telling the reader why these plots are significant (rather than just a description) would help link it back to the contributions.
* Minor: Related to presentation is the use of citations and acronyms. It seems like many acronyms are not defined, e.g. AFFD, AFJL, and QK. Also, citations should not read “While [11] developed…”. It should be “ While X et al. 2024 [11] developed…”. There is a command for this in Latex.

**Questions:**

*  Which components of the 6 contributions in the introduction are the main contributions of the paper?
* Just above equation (2), “A sketch of the HVP can be obtained as…”: Is this a contribution of the paper, or is there a reference to this available?
* In practice, which of the algorithms should one use and why?
* Could the authors define what the meaning of an influence score is for this paper?

**Limitations:**

The authors focus on scaling their approach to GPUs and TPUs and adequately talk about the memory constraints and improvements made by each approach. No code was included with the paper.

---

> ### Author Rebuttal · Authors · 2024-08-05
>
> We thank the reviewer for taking the time to read the paper, give feedback on the presentation and appreciating the theoretical part of the paper.
>
> **Answers to Questions**
>
> ```Which components of the 6 contributions in the introduction are the main contributions of the paper?```: The main contributions are 1-3 (lines 45-54). The other 3 bullet points in the intro (59-68) are novel insights enabled by contributions 1-3.
>
> ```In practice, which of the algorithms should one use and why?``` The choice depends on the target dimension and hardware, with AFFD excelling in accuracy for smaller dimensions, and QK/AFJL offering speed advantages on TPUs and GPUs, respectively. Let us look at this more closely. If the target dimension $D$ is constrained (<4k), we recommend AFFD (see Sec 5.2) for its superior accuracy. For larger $D$, QK on TPU and AFJL (with the FFT preconditioner) on GPU V100 are the fastest options. Refer to Table 2 for QK on TPU performance. For AFJL, the vanilla version takes 82ms (Table 2), and the FFT preconditioner provides a 64\% speed-up on top of that (Table 3). We believe our analysis offers a valuable set of adaptable options rather than a single ``best'' algorithm.
>
> ```Could the authors define what the meaning of an influence score is for this paper?``` We define influence score as the gradient inner product $\nabla_{\theta}L(\theta,x) \cdot \nabla_{\theta}L(\theta,z)$ (l279). We focus on this for three reasons:
>
> 1. Practicality: In the short term, it correlates with loss changes when adjusting the weighting of specific data points [Schioppa et al., https://arxiv.org/pdf/2305.16971 ].
> 1. Foundation for Advanced Methods: It serves as a building block for methods utilizing Hessian pre-conditioners or relying on gradient sketches across
> multiple model runs [TRAK, https://arxiv.org/pdf/2303.14186 ].
> 1. Clarity: The definition is straightforward and avoids additional hyper-parameters to tune (e.g., number of models or size of removed datasets in [TRAK]).
>
>
> **Clarifications related to weaknesses**
>
> We hope to address here additional concerns raised in the review.
>
> ```Minor: Figure 1 is hard to read and is too small ... There is a command for this in Latex.```: We commit to incorporate these suggestions to improve clarity and presentation.
>
>  ```Since the paper is proposing three different algorithms along with three different applications... It is a challenge to cover all of the aforementioned points in enough detail in a 9 page paper.```: We acknowledge the challenge of covering this material comprehensively in a 9-page paper. We believe there is value in presenting both (1) new theory/algorithms and (2) their applications to LLMs. Our strategy has been to split theory (Sec 3) from applications (Sec 4 & 5); each subsection of Sec 5 corresponds to a different application, so the reader can directly jump to the parts they are more interested in. We also aimed to provide sufficient references for interested readers to delve deeper. Additionally, we have included supplementary material in the Appendix. We welcome suggestions for further content to enhance the Appendix.
>
> ```No code was included with the paper.```:  Appendix B contains a step-by-step tutorial with Python code.

---

> > ### Comment · Reviewer_9kYE · 2024-08-12
> > **Thanks for the Rebuttal**
> >
> > In light of the additional results and the comments above, I will increase my score. I am still concerned with the overall presentation, but I am hopeful the authors will be able to adjust this.

---

### Official Review · Reviewer_NhaV · 2024-07-12

**Soundness:** 3
**Presentation:** 3
**Contribution:** 3
**Rating:** 6
**Confidence:** 3

**Summary:**

This paper introduces new methods for sketching high-dimensional gradients and HVPs. These are important building blocks for tools like training data attribution and Hessian spectrum analysis. Their methods introduces both empirical and theoretical improvements over prior methods, and are used to demonstrate new insights about properties of pre-trained LMs.

**Strengths:**

- Clearly written and organized
- Addresses a practically relevant and timely technical problem (projecting high-dim gradients and HVPs)
- Provides interesting new insights (e.g., that intrinsic dimension is not that low for LMs)

**Weaknesses:**

Nothing major, but:
- Misses some important related work ([1], [2])
	- but gives good coverage otherwise
- Gradient inner product is not a good TDA estimate (many papers have found this now, e.g, [1][3])
	- Estimating inner product is fine, but be clear that's the goal then

[1] TRAK: Attributing model behavior at scale https://arxiv.org/abs/2303.14186
[2] Faithful and Efficient Explanations for Neural Networks via Neural Tangent Kernel Surrogate Models https://arxiv.org/abs/2305.14585
[3] Training Data Attribution via Approximate Unrolled Differentiation https://arxiv.org/abs/2405.12186

**Questions:**

- Would benefit from better exposition of the different methods in Section 3. Some type of figure would be more useful to get intuition for differences between different methods (it's a bit hard to keep track with so many different symbols). Happy to revisit if there's a newer version.
- Some missing text, e.g., there seems to be no transition at the start of S4

---

> ### Author Rebuttal · Authors · 2024-08-06
>
> We sincerely thank the reviewer for their insightful feedback and positive assessment of our work. We commit to include [1 & 2] suggested in the review to the Related Work.
>
> **Answers to Questions**
>
> ```Would benefit from  ... there's a newer version.```:  We acknowledge the need for improved clarity in Section 3. While we cannot upload a revised paper during the rebuttal phase, we have included a proposed diagram (Figure 2) in the rebuttal PDF. We believe this visual aid will significantly enhance understanding of the various methods. We would greatly appreciate it if you could take a moment to review the diagram.
>
> ```Some missing text, ... at the start of S4```: Thank you for pointing this out. We commit to ensure that a smooth transition is added at the beginning of Section 4.
>
> **A clarification on Weaknesses**
>
> ```Gradient inner product is not a good TDA estimate```: We agree that high correlation with full gradient dot products alone does not guarantee optimal TDA performance in the long time range. However, it serves as a practical metric for short-term time evaluations and is a foundational component in more computationally demanding methods like TRAK. For instance, in the short time range, gradient dot products correlate with loss changes and are useful for example selection in error correction [Schioppa et al. https://arxiv.org/pdf/2305.16971 ]. Notably, TRAK itself relies on accurate gradient sketches as measured by dot products [TRAK, https://arxiv.org/pdf/2303.14186, C.2], with the authors stating: *as long as we preserve the inner products to sufficient accuracy, the resulting system has approximately the same evolution as the original one.* If layer selection is employed for computational efficiency, any adverse impact on the estimation of gradient inner products would have consequences for TDA even in the long run. We commit to revise the paper to make it more transparent that our method goal is to estimate inner products.

---

### Official Review · Reviewer_kKVg · 2024-07-13

**Soundness:** 3
**Presentation:** 2
**Contribution:** 3
**Rating:** 6
**Confidence:** 4

**Summary:**

Gradient information is useful for various tasks such as training data attribution and intrinsic dimension analysis, but often suffers from huge compute/memory costs, limiting its practical utility. This paper focuses on several popular sketching approaches (e.g., FJL, FFD), points out lookup-based memory as the major bottleneck in enabling efficient computations on modern accelerators, and proposes the improvement using the Kronecker products with theoretical guarantees. Their scalable sketching algorithms are applied to downstream applications including training data attribution, intrinsic dimension estimation, and Hessian analysis, and provide various useful insights, some of which are contradictory to existing common beliefs.

**Strengths:**

I agree that gradient information is useful in understanding important characteristics of neural networks, and sketching is a promising approach to reduce the cost of storing/computing gradient information. Indeed, (random) low-rank gradient projection has been studied in several prior research (e.g., Arnoldi IF, TRAK), with similar downstream applications in mind. That being said, I believe one of the main contributions of this work is in improving efficiency of existing sketching algorithms including FJL and FFD. Their solution of using the Kronecker product is, in my opinion, a smart strategy. Furthermore, their theoretical analyses are very interesting and make their arguments more rigorous. With scalable sketching algorithms, they were able to scale various analyses that were only applicable to small-scale networks to much larger networks, and provided many useful insights. For instance, many people have previously observed that full fine-tuning often achieves better performance than LoRA fine-tuning in LLMs, and the intrinsic dimension analysis in this paper can experimentally corroborates such observation. Overall, I believe the proposed algorithms would be valuable tools for understanding various phenomena in neural networks at scale.

**Weaknesses:**

In overall, I believe this paper is a strong paper *if* I limit the scope to gradient sketching. However, I am confused by several arguments.

1. TDA (Sec 5.1): The authors claimed that layer-selection for TDA is inadvisable by pointing out the low (Pearson) correlation with the full gradient dot product between (x, z) pairs. While a high correlation with full gradient dot products could indicate accurate sketching, it doesn't necessarily imply better TDA performance. There are better options for TDA evaluation such as linear data modeling score from TRAK, mislabel detection, data subset selection, brittleness tests, etc.

2. Sketching for HVP: While the authors cited matrix sketching for dealing with higher-order gradient sketching, I am unsure if theoretical guarantees apply in this case. For instance, if we adopt the Fisher information approximation of the Hessian, we can understand the Hessian as the gradient covariance matrix. If the vector in HVP is some sort of gradients, we can understand HVP, especially in influence functions, as a whitening operation that makes all components equally important. In this case, information loss from sketching is not negligible, and the general sketching argument (ie Eq(1)) doesn't really hold?

3. Clarity: I find notations in the paper to be a bit confusing. For instance, in Eq. (2, 3, 6), they used respectively P, F, \Phi to indicate the sketching process.

**Questions:**

1. I wonder how easy it is to combine proposed sketching algorithms with (data-parallel) distributed computing. While I appreciate general efficiency improvements in AFJL and AFFD, I believe distributed computing becomes necessary at some point for scaling these analyses to large-scale networks and datasets. Can you enable distributed computing simply by wrapping the model or do you need special implementation tricks?

2. How are ground-truth eigenvalues are obtained in Table 1? If those eigenvalues are not exact (e.g. approximated using some iterative methods), then being close to approximate eigenvalues do not necessarily imply better accuracy?

**Limitations:**

The authors have appropriately discussed limitations.

---

> ### Author Rebuttal · Authors · 2024-08-05
>
> We thank the reviewer for the insightful comments, taking the time to read the paper, and finding the insights provided by our sketching algorithms valuable.
>
> **Answers to Specific Questions**
>
> 1. We focused on models that don't require partitioning weights across devices. For larger models, sketching can be applied to individual partitions. We commit to including an example of lifting single-device code to multi-device code in JAX in Appendix B. Specifically, if the JAX model is sharded using jit with sharding specifications, shard-map can lift sketching algorithms to a sharded version.
>
> 2. We follow [11, Krishnan et al.], as exact eigenvalues are infeasible due to model size. This estimation is accurate, with error [11, eq(7)] decreasing exponentially in the number of iterations.
>
> **Clarifications related to weaknesses**
>
> We hope these replies help to clarify points raised in the weaknesses.
>
> 1. ``` TDA (Sec 5.1): The authors  ... brittleness tests, etc.```:  We agree that high correlation with full gradient dot products doesn't measure TDA performance in the long time range; however it is a practical metric in the short time range and a building block of more computationally intensive methods like TRAK. For example, in the short time range, gradient dot products correlate with loss changes and are relevant to select examples for error correction [Schioppa et al. https://arxiv.org/pdf/2305.16971 ]. Evaluating with LDS from TRAK would introduce more hyperparameters and computation (>50 models on different subsets of the data);  TRAK itself relies on good gradient sketches measured by dot products [TRAK, https://arxiv.org/pdf/2303.14186, C.2], and the TRAK authors  make the following point: *as long as we preserve the inner products to sufficient accuracy, the resulting system has approximately the same evolution as the original one [they refer to tracing the training dynamics, our note]. This justifies replacing the gradient features with their random projections*
>
> 2. ```Sketching for HVP: ... (ie Eq(1)) doesn't really hold?```: Theoretical guarantees for matrix sketching apply here. The Hessian has limited bulk [11, Krishnan et al.], exploited by Arnoldi-based influence functions [21, Schioppa et al. ]. Sketching provides guarantees for approximating matrix spectrums [22, Swartworth et al.] and solving linear systems [20, Sarlos, Thm 12].
>
> 3. ```Clarity: I find notations... to indicate the sketching process.```: Would replacing P, F, and $\Phi$ with $\Phi$ throughout the paper address this concern?

---

> > ### Comment · Reviewer_kKVg · 2024-08-11
> > **Response**
> >
> > > We focused on models that don't require partitioning weights across devices. For larger models, sketching can be applied to individual partitions. We commit to including an example of lifting single-device code to multi-device code in JAX in Appendix B.
> >
> > Thanks. This would be helpful.
> >
> > > We follow [11, Krishnan et al.], as exact eigenvalues are infeasible due to model size. This estimation is accurate, with error [11, eq(7)] decreasing exponentially in the number of iterations.
> >
> > Thanks for the answer.
> >
> > > Theoretical guarantees for matrix sketching apply here. The Hessian has limited bulk [11, Krishnan et al.], exploited by Arnoldi-based influence functions [21, Schioppa et al. ]. Sketching provides guarantees for approximating matrix spectrums [22, Swartworth et al.] and solving linear systems [20, Sarlos, Thm 12].
> >
> > I agree with the authors that the theoretical guarantees for sketching and apply, **if** we separately consider the Hessian and the gradient. However, when sketching is applied to both the Hessian and train/test gradients simultaneously in influence computations, there would be an inherent information loss due to the whitening effect of the inverse Hessian---all components in gradients become equally important from the whitening effect of the inverse Hessian.
> >
> > > TRAK authors make the following point: as long as we preserve the inner products to sufficient accuracy, the resulting system has approximately the same evolution as the original one [they refer to tracing the training dynamics, our note]. This justifies replacing the gradient features with their random projections
> >
> > This is true if we only consider the short horizon. In the long horizon, however, small components that contribute minimally to naive dot product may still have a non-negligible influence on the overall training dynamics.
> >
> > > Would replacing P, F, and  with  throughout the paper address this concern?
> >
> > I believe this would improve the readability.
> >
> > Considering all these, I am willing to increase my score to 6. Thanks again for your comment.

---

### Author Rebuttal · Authors · 2024-08-06

We extend our gratitude to the reviewers for their insightful comments and suggestions. We particularly appreciate the encouraging feedback, acknowledging the practical relevance and timeliness of our work, as well as its potential interest to the NeurIPS community.

In this rebuttal, we would like to highlight two key points: the use of gradient dot products and a comparison to the dense projection implementation in TRAK [ https://arxiv.org/pdf/2303.14186 ].

**Gradient Dot Products**: We concur that high correlation with full gradient dot products may not be the definitive measure of long-term TDA performance; however it is a practical metric in the short time range and a building block of more computationally intensive methods like TRAK. For example, in the short time range, gradient dot products correlate with loss changes and are relevant to select examples for error correction [Schioppa et al. https://arxiv.org/pdf/2305.16971 ]. Evaluating with LDS from TRAK would introduce more hyperparameters and computation (>50 models on different subsets of the data); TRAK itself relies on accurate gradient sketches, as measured by dot products, and the authors emphasize that preserving inner products to sufficient accuracy results in a gradient-descent system that approximately preserves the same evolution
as the one corresponding to model re-training [TRAK, https://arxiv.org/pdf/2303.14186, C.2].

**Random projections like in TRAK**: While TRAK avoids materializing the full random projection, it still requires the temporary materialization of chunks of the dense projection. This leads to two significant drawbacks: (1) runtime scales linearly with the target dimension (memory traded off with compute), and (2) specialized kernels are necessary for efficient implementation, with unclear applicability to TPUs. Our attempts to implement a competitive version using pure JAX were unsuccessful due to the lack of control over memory allocation and placement. We have included a plot (Figure 1) and a table (Table 1) in the rebuttal PDF demonstrating the linear runtime growth of TRAK compared to the constant runtimes of AFFD and QK. The implementation challenges of TRAK are further highlighted by the fact that our Triton implementation does not outperform the original CUDA kernel released by the TRAK authors, underscoring the difficulty of efficiently implementing random projections in chunks.

---

### Decision · Program_Chairs · 2024-09-25

**Decision:**

Accept (poster)

**Comment:**

The work is interesting, novel, albeit limited to gradient sketching and perhaps addresses a sub-community of NeurIPS. With this, all reviewers (and the AC) appreciate the contribution. The work have had an elaborate discussion between the authors and reviewers and we are looking forward to see their results reflected in the final submission.